# Retrieval-Guided Reinforcement Learning for Boolean Circuit Minimization

**Animesh Basak Chowdhury**[1]    **Marco Romanelli**[1]    **Benjamin Tan**[2]

**Ramesh Karri**[1]    **Siddharth Garg**[1]

[1] New York University    [2] University of Calgary

{abc586,mr6582,rkarri,nyu}@nyu.edu, {benjamin.tan1}@ucalgary.ca

## Abstract

Logic synthesis, a pivotal stage in chip design, entails optimizing chip specifications encoded in hardware description languages like Verilog into highly efficient implementations using Boolean logic gates. The process involves a sequential application of logic minimization heuristics ("synthesis recipe"), with their arrangement significantly impacting crucial metrics such as area and delay. Addressing the challenge posed by the broad spectrum of design complexities — from variations of past designs (e.g., adders and multipliers) to entirely novel configurations (e.g., innovative processor instructions) — requires a nuanced 'synthesis recipe' guided by human expertise and intuition. This study conducts a thorough examination of learning and search techniques for logic synthesis, unearthing a surprising revelation: pre-trained agents, when confronted with entirely novel designs, may veer off course, detrimentally affecting the search trajectory. We present ABC-RL, a meticulously tuned $\alpha$ parameter that adeptly adjusts recommendations from pre-trained agents during the search process. Computed based on similarity scores through nearest neighbor retrieval from the training dataset, ABC-RL yields superior synthesis recipes tailored for a wide array of hardware designs. Our findings showcase substantial enhancements in the Quality-of-result (QoR) of synthesized circuits, boosting improvements of up to 24.8% compared to state-of-the-art techniques. Furthermore, ABC-RL achieves an impressive up to 9x reduction in runtime (iso-QoR) when compared to current state-of-the-art methodologies.

## 1 Introduction

Modern chips are designed using sophisticated electronic design automation (EDA) algorithms that automatically convert logic functions expressed in a hardware description language (HDL) like Verilog to a physical layout that can be manufactured at a semiconductor foundry. EDA involves a sequence of steps, the first of which is *logic synthesis*. Logic synthesis converts HDL into a low-level "netlist" of Boolean logic gates that implement the desired function. A netlist is a graph whose nodes are logic gates (*e.g.*, ANDs, NOTs, ORs) and whose edges represent connections between gates. Subsequent EDA steps like physical design then place gates on an x-y surface and route wires between them. As the *first* step in the EDA flow, any inefficiencies in this step, *e.g.*, redundant logic gates, flow down throughout the EDA flow. Thus, the quality of logic synthesis—the area, power and delay of the synthesized netlist—is *crucial* to the quality of the final design (Amarú et al., 2017).

As shown in Fig. 1, state-of-art logic synthesis algorithms perform a sequence of functionality-preserving transformations, *e.g.*, eliminating redundant nodes, reordering Boolean formulas, and streamlining node representations, to arrive at a final optimized netlist Yang et al. (2012); Mishchenko et al. (2006); Riener et al. (2019); Yu (2020); Neto et al. (2022). A specific sequence of transformations is called a "**synthesis recipe.**" Typically, designers use experience and intuition to pick a "good" synthesis recipe from the solution space of all recipes and iterate if the quality of result is poor. This manual process is costly and time-consuming, especially for modern, complex chips. Further, the design space of synthesis recipes is large. ABC (Brayton & Mishchenko, 2010), a state-of-art open-source synthesis tool, provides a toolkit of 7 transformations, yielding a design space of $10^7$ recipes (assuming 10-step recipes). A growing body of work has sought to leverage machine learning

and reinforcement learning (RL) to automatically identify high-quality synthesis recipes (Yu et al., 2018; Hosny et al., 2020; Zhu et al., 2020; Yu, 2020; Neto et al., 2022; Chowdhury et al., 2022), showing promising results.

Prior work in this area falls within one of two categories. The first line of work (Yu, 2020; Neto et al., 2022) proposes efficient *search* heuristics, Monte-Carlo tree search (MCTS) in particular, to explore the solution space of synthesis recipes for a given netlist. These methods train a policy agent during MCTS iterations, but the agent is initialized from scratch for a given netlist and does *not* learn from past data, *e.g.*, repositories of previously synthesized designs abundant in chip design companies, or sourced from public repositories (Chowdhury et al., 2021; Thakur et al., 2023).

To leverage experience from past designs, a second line of work seeks to *augment search with learning*. Chowdhury et al. (2022) show that a predictive QoR model trained on past designs with simulated annealing-based search can outperform prior search-only methods by as much as 20% in area and delay. The learned model replaces time-consuming synthesis runs—a single synthesis run can take around 10.9 minutes in our experiments—with fast but approximate QoR estimates from a pre-trained deep network.

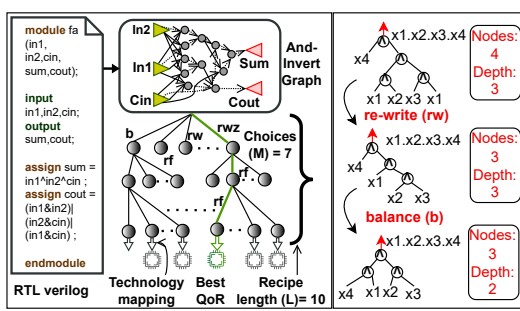

**Figure 1:** (Left) A hardware design in Verilog is first transformed into an and-inverter-graph (AIG), i.e., a netlist containing only AND and NOT gates. Then a sequence of functionality-preserving transformations (here, picked from set {**rw**, **rwz**, ..., **b** }) is applied to generate an optimized AIG. Each such sequence is called a synthesis recipe. The synthesis recipe with the best quality of result (QoR) (e.g., area or delay) is shown in green. (Right) Applying **rw** and **b** to an AIG results results in an AIG with fewer nodes and lower depth.

**Motivational Observation.** Given the tree-structured search space (see Fig. 2), we begin by building and evaluating a baseline solution that: 1) pre-trains an offline RL agent on a dataset of past designs; and then 2) performs RL-agent guided MCTS search over synthesis recipe space for new designs. Although details vary, this strategy has been successful in other tree-search

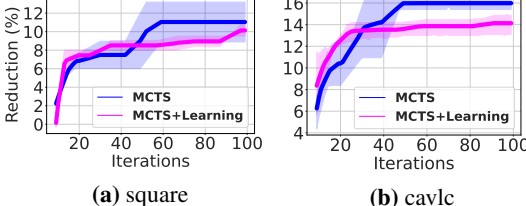

(a) square    (b) cavlc

**Figure 2:** Reduction in area-delay product (greater reduction is better) over search iterations for a pure search strategy (MCTS) and search augmented with learning (MCTS+Learning). Learning an offline policy does not help in both cases.

problems, most prominently in AlphaGo (Silver et al., 2016) and AlphaZero Silver et al. (2017). Interestingly, we found that although the agent learned on past data helps slightly on *average*, on 11 out of 20 designs, the baseline strategy underperformed simple MCTS search (see Table 2 for detailed results). Fig. 2 shows two examples: in both, pure MCTS achieves better solutions faster than learning augmented MCTS. Ideally, we seek solutions that provide consistent improvements over search-only methods by leveraging past data.

One reason for poor performance is the large diversity of netlists in logic synthesis benchmarks. Netlists vary significantly in size (100-46K nodes) and function. The EPFL benchmarks, for example, partition netists into those that perform arithmetic functions (*e.g.*, adders, dividers, square-roots) and control functions (*e.g.*, finite state machines, pattern detectors, *etc.*) because of large differences in their graph structures. In practice, while designers often reuse components from past designs, they frequently come up with new designs and novel functions. For netlists that differ significantly from those in the training dataset, pre-trained agents hurt by diverting search towards suboptimal recipes.

**Overview of Approach.** We propose ABC-RL, a new retrieval guided RL approach that adaptively tunes the contribution of the pre-trained policy agent in the online search stage depending on the input netlist. ABC-RL computes a tuning factor $\alpha \in [0, 1]$ by computing a similarity score of the input netlist and its nearest neighbor retrieved from the training dataset. Similarity is computed on graph neural network (GNN) features learned during training. If the new netlist is identical to one in the training dataset, we set $\alpha = 0$, and *only* the pre-trained agent is used to output the synthesis recipe. Conversely, when $\alpha = 1$ (i.e., the netlist is novel), the pre-trained agent is ignored and ABC-RL

defaults to a search strategy. Real-world netlists lie in between these extremes; accordingly ABC-RL modulating the relative contributions of the pre-trained agent and pure MCTS search to the final result. We make careful architectural choices in our implementation of ABC-RL, including the choice of netlist and synthesis recipe encoders and state-space representation. They are described in Section 2.3. Although our main contribution is ABC-RL, the MCTS+Learning baseline (i.e., without retrieval) has not been evaluated for logic synthesis. Our evaluations highlight its benefits and drawbacks.

**Snapshot of Results and Key Contributions.** Across three common logic synthesis benchmark suites ABC-RL consistently outperforms prior SOTA ML-based logic synthesis solutions, our own MCTS+Learning baseline, *and* an MCTS+Learning solution for chip placement Mirhoseini et al. (2021), a different EDA problem, adapted to logic synthesis. ABC-RL achieves upto 24.8% geo. mean improvements in QoR (here, area-delay product) over SOTA. At iso-QoR, ABC-RL reduces synthesis runtime by up to $9\times$.

In summary, our key contributions are:

- We propose ABC-RL, a new retrieval-guided RL approach for logic synthesis that learns from past historical data, i.e., previously seen netlists, to optimize the QoR for a new netlist at test time. In doing so, we identify distribution shift between training and test data as a key problem in this domain, and show that a baseline strategy that augments MCTS with a pre-trained policy agent Silver et al. (2016) fails to improve upon pure MCTS search.
- To address these concerns, we introduce new lightweight retrieval mechanism in ABC-RL that uses the similarity score between the new test netlist and its nearest neighbor in the training set. This score modulates the relative contribution of pre-trained RL agent using a modulation parameter $\alpha$, down-weighting the learned policy depending on the novelty of the test netlist.
- We make careful architectural choices for ABC-RL's policy agent, including a new transformer based synthesis encoder, and evaluate across three of common logic synthesis benchmarks. ABC-RL consistently outperforms prior SOTA ML for logic synthesis methods on QoR and runtime, as also a recent ML for chip-placement method (Mirhoseini et al., 2021) adapted to our problem setting. Ablation studies establish the importance of $\alpha$-tuning to ABC-RL.

We now describe ABC-RL, starting with a precise problem formulation and background.

## 2 PROPOSED APPROACH

### 2.1 PROBLEM STATEMENT AND BACKGROUND

We begin by formally defining the logic synthesis problem using ABC (Brayton & Mishchenko, 2010), the leading open-source synthesis tool, as an exemplar. As shown in Figure 1, ABC first converts the Verilog description of a chip into an unoptimized And-Invert-Graph (AIG) $G_0 \in \mathcal{G}$, where $\mathcal{G}$ is the set of all directed acyclic graphs. The AIG represents AND gates as nodes, wires/NOT gates as solid/dashed edges, and implements the same Boolean function as the input Verilog. Next, ABC performs a series of functionality-preserving transformations on $G_0$. Transformations are picked from a finite set of $M$ actions, $\mathcal{A} = \{\mathrm{rf}, \mathrm{rm}, \ldots, \mathrm{b}\}$. For ABC, $M = 7$. Applying an action on an AIG yields a new AIG as determined by the synthesis function $\mathbf{S} : \mathcal{G} \times \mathcal{A} \to \mathcal{G}$. A synthesis recipe $R \in \mathcal{A}^L$ is a sequence of $L$ actions that are applied to $G_0$ in order. Given a synthesis recipe $P = \{a_0, a_1, \ldots, a_{L-1}\}$ $(a_i \in \mathcal{A})$, we obtain $G_{i+1} = \mathbf{S}(G_i, a_i)$ for all $i \in [0, L-1]$ where $G_L$ is the final *optimized* AIG. Finally, let $\mathbf{QoR} : \mathcal{G} \to \mathbb{R}$ measure the quality of graph $G$, for instance, its inverse area-delay product (so larger is better). Then, we seek to solve this optimization problem:

$$\underset{P \in \mathcal{A}^L}{\mathrm{argmax}} \, \mathbf{QoR}(G_L), \ s.t. \ G_{i+1} = \mathbf{S}(G_i, a_i) \ \forall i \in [0, L-1]. \tag{1}$$

We now discuss ABC-RL, our proposed approach to solve this optimization problem. In addition to $G_0$, the AIG to be synthesized, we will assume access to a training set of AIGs to aid optimization.

### 2.2 BASELINE MCTS-BASED OPTIMIZATION

The tree-structured solution space for logic synthesis motivated prior work (Yu, 2020; Neto et al., 2022) to adopt an MCTS-based search strategy that we briefly review here. A state $s$ here is the

current AIG after $l$ transformations. In a given state, any action $a \in \mathcal{A}$ can be picked as described above. Finally, the reward $\mathbf{QoR}(G_L)$ is delayed to the final synthesis step. In iteration $k$ of the search, MCTS keeps track of two functions: $Q^k_{MCTS}(s, a)$ which is measure the "goodness" of a state action pair, and $U^k_{MCTS}(s, a)$ which represents upper confidence tree (UCT) factor that encourages exploration of less visited states and actions. The policy $\pi^k_{MCTS}(s)$ balances exploitation against exploration by combining the two terms. Further details are in §A.3.

$$\pi^k_{MCTS}(s) = \underset{a \in \mathcal{A}}{\operatorname{argmax}} \left( Q^k_{MCTS}(s, a) + U^k_{MCTS}(s, a) \right). \quad (2)$$

## 2.3 Proposed ABC-RL Methodology

We describe our proposed solution in two steps. First, we describe MCTS+Learning, which builds similar principles as Silver et al. (2016) by training an reinforcement learning (RL) policy agent on prior netlists to guide Monte Carlo tree search (MCTS) search, highlighting how prior work is adapted to the logic synthesis problem. Then, we describe our full solution, ABC-RL, that uses novel retrieval-guided augmentation to significantly improve MCTS+Learning.

**MCTS+Learning:** As noted, we use a dataset of $N_{tr}$ training circuits to learn a policy agent $\pi_\theta(s, a)$ that outputs the probability of taking action $a$ in state $s$ by approximating the pure MCTS policy on the training set. We first describe our state-space representation and policy network architecture.

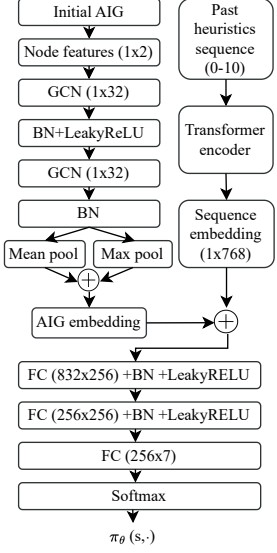

**Figure 3:** Policy network architecture. GCN: Graph convolution network, BN: Batch normalization, FC: Fully connected layer

**State-space and policy network architecture:** We encode state $s$ as as a two-tuple of the *input* AIG, $G_0$, and sequence of $l \leq L$ synthesis actions, i.e., $A_l = \{a_0, a_1, \ldots, a_l\}$ taken so far. Because the two inputs are in different formats, our policy network has two parallel branches that learn embeddings of AIG $G_0$ and partial recipe.

For the AIG input, we employ a 3-layer graph convolutional network (GCN) Kipf & Welling (2016) architecture to obtain an embedding $h_{G_0}$. We use LeakyRELU as the activation function and apply batch normalization before each layer. (See appendix §B.1 for details.) In contrast to prior work that directly encodes recipes Chowdhury et al. (2022), we use a simple *single* attention layer BERT transformer architecture Devlin et al. (2018) to compute partial recipe embeddings, $h_{A_l}$, which we concatenate with AIG embeddings. We make this choice since partial synthesis recipes are variable length, and to better capture contextual relationships within a sequence of actions. Ablation studies demonstrate the advantages of this approach. The final embedding is a concatenation on the AIG and partial synthesis recipe embeddings.

**RL-agent training:** With the policy network in place, policy $\pi_\theta(s, a)$ is learned on a training dataset of past netlists using a cross-entropy loss between the learned policy and the MCTS policy over samples picked from a replay buffer. The learned policy is used during *inference* to bias the upper confidence tree (UCT in Eq. 2) of MCTS towards favorable paths by computing a new $U^{*k}_{MCTS}(s, a)$. For completeness, we outline the pseudocode for RL-training in the appendix (Algorithm 1).

$$U^{*k}_{MCTS}(s, a) = \pi_\theta(s, a) \cdot U^k_{MCTS}(s, a). \quad (3)$$

## 2.4 Retrieval-guided Logic Synthesis (ABC-RL)

As we noted, hardware designs frequently contain both familiar and entirely new components. In the latter case, our results indicate that the learned RL-agents can sometimes *hurt* performance on novel inputs by biasing search towards sub-optimal synthesis recipes. In ABC-RL, we introduce a new term $\alpha \in [0, 1]$ in Equation 3 that strategically weights the from pre-trained agents contribution, completely turning it off when $\alpha = 1$ (novel circuit) and defaulting to the baseline approach when $\alpha = 0$.

**Figure 4:** ABC-RL flow: Training the agent (left), setting temperature $T$ and threshold $\delta_{th}$ (mid) and Recipe generation at inference-time (right)

**(1) Similarity score computation:** To quantify novelty of a new netlist, $G_0$, at test-time, we compute a similarity score with respect to its nearest neighbor in the training dataset $D_{tr} = \{G_0^{tr}, \ldots, G_{N_t r}^{tr}\}$. To avoid expenseive nearest neighbor queries in the graph space, for instance, via sub-graph isomorphisms, we leverage the graph encodings, $h_G$ already learned by the policy agent. Specifically, we output the smallest cosine distance ($\Delta_{cos}(h_{G_1}, h_{G_2}) = 1 - \frac{h_{G_1} \cdot h_{G_2}}{|h_{G_1}||h_{G_2}|}$) between the test AIG embedding and all graphs in the training set: $\delta_{G_0} = \min_i \Delta_{cos}(h_{G_0}, h_{G_i^{tr}})$.

**(2) Tuning agent's recommendation during MCTS:** To modulate the balance between the prior learned policy and pure search, we update the prior UCT with $\alpha \in [0, 1]$ as follows:

$$U_{MCTS}^{*k}(s, a) = \pi_\theta(s, a)^{(1-\alpha)} \cdot U_{MCTS}^k(s, a), \tag{4}$$

and $\alpha$ is computed by passing similarity score, $\delta_{G_0}$, through a sigmoid function, $\alpha = \sigma_{\delta_{th}, T}(\delta_{G_0})$, defined as $\sigma_{\delta_{th}, T}(z) = \frac{1}{1 + e^{-\frac{z - \delta_{th}}{T}}}$ with threshold ($\delta_{th}$) and temperature ($T$) hyperparameters.

Eq.4 allows $\alpha$ to smoothly vary in $[0, 1]$ as intended, while threshold and temperature hyperparameters control the shape of the sigmoid. Threshold $\delta_{th}$ controls how close new netlists have to be to the training data to be considered "novel." In general, small thresholds bias ABC-RL towards more search and less learning from past data. Temperature $\delta_{th}$ controls the transition from "previously seen" to novel. Small temperatures cause ABC-RL to create a harder thresholds between previously seen and novel designs. Both hyperparameters are chosen using validation data.

**(3) Putting it all together:** In Fig. 4, we present an overview of the ABC-RL. We begin by training an RL-agent on training dataset $D_{tr}$. Then, we use a separate held-out validation dataset to tune threshold, $\delta_{th}$, and temperature, $T$, by comparing wins/losses of baseline MCTS+Learning vs. ABC-RL and performing a grid-search. During inference on a new netlist $G_0$, ABC-RL retrieves the nearest neighbor from the training data, computes $\alpha$ and performs online $\alpha$-guided search using weighted recommendations from the pre-trained RL agent.

## 3 EMPIRICAL EVALUATION

### 3.1 EXPERIMENTAL SETUP

**Datasets:** We consider three datasets used by logic synthesis community: MCNC Yang (1991), EPFL arithmetic and EPFL random control benchmarks Amarú et al. (2015). MCNC benchmarks have 38 netlists ranging from 100–8K nodes. EPFL arithmetic benchmarks have operations like additions, multiplications etc. and have 1000-44K nodes. EPFL random control benchmarks have finite-state machines, routing logic and other functions with 100 –46K nodes.

| Split | Circuits |
|-------|----------|
| Train | alu2, apex3, apex5, b2, c1355, c5315, c2670, c6288, prom2, frg1, i7, i8, m3, max512, table5, adder, log2, max, multiplier, arbiter, ctrl, int2float, priority |
| Valid | apex7, c1908, c3540, frg2, max128, apex6, c432, c499, seq, table3, i10, sin, i2c |
| Test | alu4, apex1, apex2, apex4, i9, m4, prom1, b9, c880, c7552, pair, max1024 {C1-C12}, bar, div, square, sqrt {A1-A4}, cavlc, mem_ctrl, router, voter {R1-R4} |

**Table 1:** Training, validation and test splits in our experiments. Netlists from each benchmark are represented in each split. In the test set, MCNC netlists are relabeled [C1-C12], EPFL-arith to [A1-A4] and EPFL-control to [R1-R4].

**Train-test split:** We randomly split the 56 total netlists obtained from all three benchmarks into 23 netlists for training 13 for validation (11 MCNC, 1 EPFL-arith, 1 EPFL-rand) and remaining

20 for test (see Table 1). We ensure that netlists from each benchmark are represented proportionally in training, validation and test data.

**Optimization objective and metrics:** We seek to identify the best $L = 10$ synthesis recipes. Consistent with prior works Hosny et al. (2020); Zhu et al. (2020); Neto et al. (2022), we use area-delay product (ADP) as our QoR metric. Area and delay values are obtained using a 7nm technology library post-technology mapping of the synthesized AIG. As a baseline, we compare against ADP of the *resyn2* synthesis recipe as is also done in prior work Neto et al. (2022); Chowdhury et al. (2022). In addition to ADP reduction, we report runtime reduction of ABC-RL at iso-QoR, i.e., how much faster ABC-RL is in reaching the best ADP achieved by competing methods. During evaluations on test circuits, we give each technique a total budget of 100 synthesis runs.

**Training details and hyper-parameters:** To train our RL-agents, we use He initialization He et al. (2015) for weights and following Andrychowicz et al. (2020), multiply weights of the final layer with 0.01 to prevent bias towards any one action. Agents are trained for 50 epochs using Adam with an initial learning rate of 0.01. In each training epoch, we perform MCTS on all netlists with an MCTS search budget $K = 512$ per synthesis level. After MCTS simulations, we sample $L \times N_{tr}$ ($N_{tr}$ is the number of training circuits) experiences from the replay buffer (size $2 \times L \times N_{tr}$) for training. To stabilize training, we normalize QoR rewards (Appendix C.1) and clip it to $[-1, +1]$ Mnih et al. (2015). We set $T = 100$ and $\delta_{th} = 0.007$ based on our validation data.

We performed the training on a server machine with one NVIDIA RTX A4000 with 16GB VRAM. The major bottleneck during training is the synthesis time for running ABC; actual gradient updates are relatively inexpensive. Fully training the RL-agent training took around 9 days.

**Baselines for comparison:** We compare ABC-RL with five main methods: (1) **MCTS:** Search-only MCTS Neto et al. (2022); (2) **DRiLLS:** RL agent trained via A2C using hand-crafted AIG features (not on past-training data) Hosny et al. (2020) (3) **Online-RL:** RL agent trained online via PPO using Graph Convolutional Networks for AIG feature extraction (but not on past training data) (Zhu et al., 2020); (4) **SA+Pred:** simulated annealing (SA) with QoR predictor learned from training data Chowdhury et al. (2022); and (5) **MCTS+L(earning):** our own baseline MCTS+Learning solution using a pre-trained RL-agent (Section **??**). MCTS and SA+Pred are the current state-of-the-art (SOTA) methods. DRiLLS and Online-RL are similar and under-perform MCTS. We report results versus exisitng methods for completeness. MCTS+L is new and has not been evaluated on logic synthesis. A final and sixth baseline for comparison is **MCTS+L+FT** (Mirhoseini et al., 2021) which is proposed for chip placement, a different EDA problem, but we adapt for logic synthesis. MCTS+FT is similar to MCTS+L but continues to fine-tune its pre-trained RL-agent on test inputs.

## 3.2 Experimental Results

### 3.2.1 ABC-RL Vs. SOTA

In Table 2, we compare ABC-RL over SOTA methods Hosny et al. (2020); Zhu et al. (2020); Neto et al. (2022); Chowdhury et al. (2022) in terms of percentage area-delay product (ADP) reduction (relative to the baseline *resyn2* recipe). We also report speed-ups of ABC-RL to reach the same QoR as the SOTA methods. Given the long synthesis runtimes, speed-up at iso-QoR as an equally important metric. Overall, **ABC-RL achieves the largest geo. mean reduction in ADP**, reducing ADP by 25% (smaller ADP is better) over *resyn2*. The next best method is our own MCTS+L implementation which achieves 20.7% ADP reduction, although it is only slightly better than MCTS.

ABC-RL is also **consistently the winner in all but four netlists**, and in each of these cases ABC-RL finishes second only marginally behind the winner. Interestingly, the winner in these cases is Online+RL, a method that overall has the poorest performance. Importantly, ABC-RL is consistently better than MCTS+L *and* MCTS. That is, **we show that both learning on past data and using retrieval are key to good performance.** Finally, ABC-RL is also **faster than competing methods** with a geo. mean speed-up of $3.8\times$ over SOTA. We dive deeper into specific benchmark suite to understand where ABC-RL's improvements stem from.

**MCNC benchmarks:** ABC-RL provides with substantial improvements on benchmarks such as C1 (`apex1`), C7 (`prom1`), C9 (`c880`), and C12 (`max1024`). Fig. 5 plots the ADP reductions over search iterations for MCTS, SA+Pred, and ABC-RL over four netlists from MCNC. Note from Fig. 5 that ABC-RL in most cases achieves higher ADP reductions earlier than competing methods. Thus,

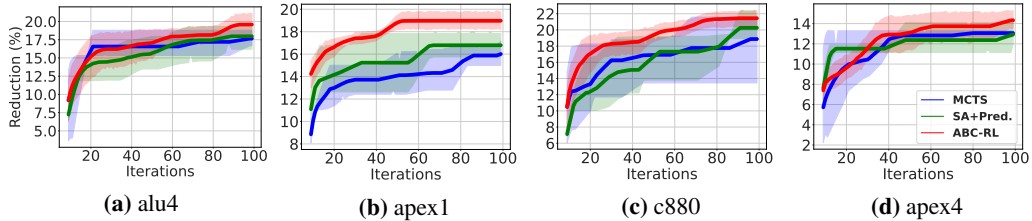

**Figure 5:** Area-delay product reduction (in %) compared to *resyn2* on MCNC circuits.

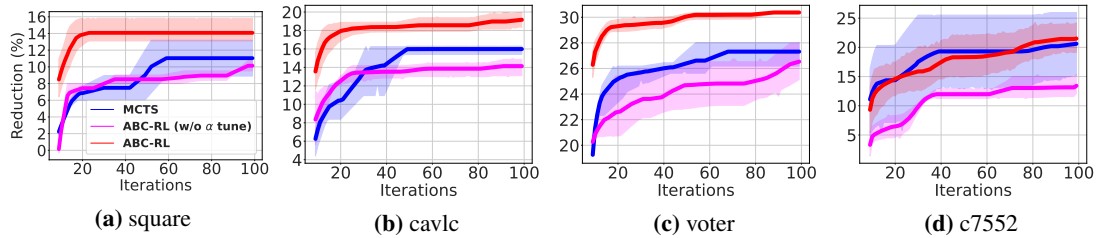

**Figure 6:** Area-delay product reduction (in %) using ABC-RL compared to MCTS+Learning.

designers can terminate search when a desired ADP is achieved. This results in run-time speedups upto $5.9\times$ at iso-QoR compared to standard MCTS. (see Appendix D.1.1 for complete results)

**Table 2:** Area-delay reduction (%) compared to *resyn2*: DRiLLS Hosny et al. (2020), Online-RL Zhu et al. (2020), SA+Pred. Chowdhury et al. (2022), MCTS Neto et al. (2022), MCTS+Learning (MCTS+L) and ABC-RL. Speed-up vs. MCTS

| Methods | ADP reduction (in %) | | | | | | | | | | | | | | | | | | | | Geo-mean |
|---------|------|------|------|------|------|------|------|------|------|------|------|------|------|------|------|------|------|------|------|------|------|
| | MCNC | | | | | | | | | | | | EPFL arith | | | | EPFL random | | | | |
| | C1 | C2 | C3 | C4 | C5 | C6 | C7 | C8 | C9 | C10 | C11 | C12 | A1 | A2 | A3 | A4 | R1 | R2 | R3 | R4 | |
| DRiLLS | 18.9 | 6.7 | 8.0 | 13.0 | 38.4 | 19.1 | 5.4 | 18.0 | 14.3 | 18.6 | 6.6 | 11.0 | 28.8 | 34.7 | 11.1 | 22.7 | 15.4 | 23.0 | 12.9 | 10.1 | 16.1 |
| Online-RL | 20.6 | 6.6 | 8.1 | 13.5 | 39.4 | 21.0 | 5.0 | 17.9 | 16.2 | 20.2 | 4.7 | 11.4 | 36.9 | 34.8 | 10.4 | 24.1 | 16.3 | 22.5 | 10.7 | 8.3 | 16.1 |
| SA+Pred. | 17.6 | 17.0 | 15.6 | 13.0 | 46.5 | 18.2 | 8.5 | 23.6 | 19.9 | 17.6 | 10.0 | 20.3 | 36.9 | 25.2 | 8.2 | 21.1 | 16.8 | 21.5 | 25.7 | 26.2 | 19.7 |
| MCTS | 17.1 | 15.9 | 13.1 | 13.0 | 46.9 | 14.9 | 6.5 | 23.2 | 17.7 | 20.5 | 13.1 | 19.7 | 25.4 | 46.0 | 10.7 | 18.7 | 15.9 | 21.6 | 21.6 | 27.1 | 19.8 |
| MCTS+L | 17.0 | 19.6 | 16.9 | 12.5 | 46.9 | 13.9 | 10.1 | 24.1 | 17.1 | 16.8 | 8.1 | 19.5 | 36.9 | 55.9 | 10.3 | 22.7 | 15.8 | 24.1 | 38.9 | 26.9 | 20.7 |
| **ABC-RL** | 19.9 | 19.6 | 16.8 | 15.0 | 46.9 | 19.1 | 12.1 | 24.3 | 21.3 | 21.1 | 13.6 | 21.6 | 36.9 | 56.2 | 14.0 | 23.8 | 19.8 | 30.2 | 38.9 | 30.0 | 25.3 |
| Iso-QoR Speed-up | 1.9x | 5.9x | 1.8x | 1.6x | 1.2x | 1.3x | 3.2x | 4.1x | 2.2x | 1.2x | 0.9x | 1.8x | 3.7x | 9.0x | 4.2x | 8.3x | 5.0x | 6.4x | 3.1x | 5.7x | 3.8x |

**EPFL benchmarks:** ABC-RL excels on 7 out of 8 EPFL designs, particularly demonstrating significant improvements on A3 (`square`), R1 (`cavlc`), and R3 (`mem_ctrl`) compared to prior methods Hosny et al. (2020); Zhu et al. (2020); Chowdhury et al. (2022); Neto et al. (2022). Notably, baseline MCTS+Learning performs poorly on A3 (`square`), R1 (`cavlc`), and R4 (`voter`). In Fig. 6, we illustrate how ABC-RL fine-tunes $\alpha$ for various circuits, carefully adjusting pre-trained recommendations to avoid unproductive exploration paths. Across all EPFL benchmarks, ABC-RL consistently achieves superior ADP reduction compared to pure MCTS, with a geometric ADP reduction of 28.85% over *resyn2*. This significantly improves QoR over standard MCTS and SA+Pred., by 5.99% and 6.12% respectively. Moreover, ABC-RL delivers an average of $1.6\times$ runtime speed-up at iso-QoR compared to standard MCTS Neto et al. (2022), up to $9\times$ speed-up.

### 3.2.2 BENCHMARK SPECIFIC ABC-RL AGENTS VS. SOTA

To further examine the benefits of ABC-RL in terms of netlist diversity, we train three benchmark-specific ABC-RL agents on each benchmark suite using, as before, the train-vaidation-test splits in Table 1. Although trained on each benchmark individually, evaluation of benchmark-specific agents is on the full test dataset. This has two objectives: 1) Assess how benchmark-specific agents compare against the benchmark-wide agents on their own test inputs, and 2) Study how ABC-RL's benchmark-

specific agents adapt when deployed on test inputs from other benchmarks. Benchmark-specific agents are referred to as ABC-RL+X, where X is the benchmark name (MCNC, ARITH, or RC).

**Table 3:** Area-delay reduction (in %) obtained using benchmark specific agents (MCNC, ARITH and RC). MCTS represents tree search adopted in Neto et al. (2022)

| ABC-RL agents /Methods | ADP reduction (in %) | | | | | | | | | | | | | | | | | | | | Geo. mean |
|---|---|---|---|---|---|---|---|---|---|---|---|---|---|---|---|---|---|---|---|---|---|
| | MCNC | | | | | | | | | | | | EPFL arith | | | | EPFL random | | | | |
| | C1 | C2 | C3 | C4 | C5 | C6 | C7 | C8 | C9 | C10 | C11 | C12 | A1 | A2 | A3 | A4 | R1 | R2 | R3 | R4 | |
| ABC-RL | 19.9 | **19.6** | 16.8 | **15.0** | 46.9 | 19.1 | **12.1** | **24.3** | 21.3 | 21.1 | 13.6 | 21.6 | **36.9** | **56.2** | **14.0** | 23.8 | **19.8** | **30.2** | **38.9** | **30.0** | **25.3** |
| +MCNC | **20.9** | 19.2 | **17.5** | **15.0** | **52.5** | 18.1 | 10.9 | 24.1 | **24.7** | 21.1 | **16.8** | **22.0** | 32.9 | 47.9 | 10.9 | 18.7 | 18.2 | 24.0 | 24.5 | 27.1 | 21.3 |
| +ARITH | 18.0 | 16.0 | 17.3 | 13.0 | 46.9 | **20.3** | 8.8 | 23.2 | 20.0 | **24.1** | 13.1 | 20.8 | **36.9** | 55.9 | 12.1 | **25.1** | 16.9 | 21.5 | 21.8 | 27.7 | 21.4 |
| +RC | 19.0 | 18.5 | 17.4 | 12.5 | 46.9 | 16.8 | 11.9 | 23.3 | 22.5 | 20.4 | 13.1 | 20.0 | 36.5 | 53.4 | 10.8 | 18.7 | 17.8 | 22.8 | 26.6 | 27.1 | 21.7 |
| MCTS | 17.1 | 15.9 | 13.1 | 13.0 | 46.9 | 14.9 | 6.5 | 23.2 | 17.7 | 20.5 | 13.1 | 19.7 | 25.4 | 45.9 | 10.7 | 18.7 | 15.9 | 21.6 | 21.6 | 27.1 | 19.8 |

In Table 3, we present the performance of ABC-RL using benchmark-specific agents. Notably, ABC-RL+X agents often outperform the general ABC-RL agent on test inputs from their own benchmark suites. For example, ABC-RL+MCNC outperforms ABC-RL on 7 of 12 benchmarks. In return, the performance of benchmark-specific agents drops on test inputs from other benchmarks because these new netlists are novel for the agent. Nonetheless, our benchmark-specific agents **still outperform the SOTA MCTS approach in geo. mean ADP reduction.** In fact, if except ABC-RL, each of our benchmark-specific agents would still outperform other SOTA methods including MCTS+L. These results emphasize ABC-RL's ability to fine-tune $\alpha$ effectively, even in the presence of a substantial distribution gap between training and test data.

## 3.3 ABC-RL Vs. MCTS+L+FT

In recent work, Mirhoseini et al. (2021) proposed a pre-trained PPO agent for chip placement. This problem seeks to place blocks on the chip surface so as to reduce total chip area, wire-length and congestion. Although the input to chip placement is also a graph, the graph only encodes connectivity and not functionality. Importantly, an action in this setting, *e.g.* moving or swapping blocks, is *quick*, allowing for millions of actions to be explored. In contrast, for logic synthesis, actions (synthesis steps) involve expensive functionality-preserving graph-level transformations on the entire design taking up to 5 minutes for larger designs. To adapt to new inputs, Mirhoseini et al. (2021) adopt a different strategy: they continue to fine-tune (FT) their agents as they perform search on test inputs. Here we ask if the FT strategy could work for ABC-RL instead of our retrieval-guided solution.

**Table 4:** Area-delay reduction (in %). ABC-RL−BERT is ABC-RL trained with naive synthesis encoder instead of BERT. MCTS+L+FT indicate MCTS+Learning with online fine-tuning.

| Ablation study | ADP reduction (in %) | | | | | | | | | | | | | | | | | | | | Geo. mean |
|---|---|---|---|---|---|---|---|---|---|---|---|---|---|---|---|---|---|---|---|---|---|
| | MCNC | | | | | | | | | | | | EPFL arith | | | | EPFL random | | | | |
| | C1 | C2 | C3 | C4 | C5 | C6 | C7 | C8 | C9 | C10 | C11 | C12 | A1 | A2 | A3 | A4 | R1 | R2 | R3 | R4 | |
| ABC-RL | 19.9 | 19.6 | 16.8 | 15.0 | 46.9 | 19.1 | 12.1 | 24.3 | 21.3 | 21.1 | 13.6 | 21.6 | 36.9 | 56.2 | 14.0 | 23.8 | 19.8 | 30.2 | 38.9 | 30.0 | 25.3 |
| MCTS+L+FT | 17.1 | 18.0 | 15.0 | 14.1 | 37.9 | 12.9 | 10.1 | 24.3 | 17.3 | 19.6 | 10.0 | 20.0 | 36.9 | 55.9 | 10.7 | 22.1 | 20.0 | 30.2 | 38.9 | 28.3 | 23.3 |
| ABC-RL− BERT | 17.0 | 16.5 | 14.9 | 13.1 | 44.6 | 16.9 | 10.0 | 23.5 | 16.3 | 18.8 | 10.6 | 19.0 | 36.9 | 51.7 | 9.9 | 20.0 | 15.5 | 23.0 | 28.0 | 26.9 | 21.2 |

To test this, we fine-tune ABC-RL's benchmark-wide agent during online MCTS within our evaluation budget of 100 synthesis runs. Table 4 compares ABC-RL vs. the new MCTS+L+FT approach. ABC-RL outperforms MCTS+L+FT on all but one netlist, and reduces ADP by 2.66%, 2.40% and 0.33% on MCNC, EPFL, and random control benchmarks, with 9.0% decline on C5 (`i9`).

### 3.3.1 Impact of Architectural Choices

We inspect the role of BERT-based recipe encoder in ABC-RL by replacing it with a fixed length ($L = 10$) encoder where, using the approach from (Chowdhury et al., 2022), we directly encode the synthesis commands in numerical form and apply zero-padding for recipe length less than $L$. The results are shown in Table 4. ABC-RL reduces ADP by 2.51%, 4.04% and 4.11% on MCNC, EPFL arithmetic and random control benchmarks, and upto -10.90% decline on R3 (`router`), compared to the version without BERT. This shows the importance of transformer-based encoder in extracting meaningful features from synthesis recipe sub-sequences for state representation.

## 4 RELATED WORK

**Learning-based approaches for logic synthesis:** This can be classified into two sub-categories: 1) Synthesis recipe classification (Yu et al., 2018; Neto et al., 2019) and prediction (Chowdhury et al., 2021; 2022) based approaches, and 2) RL-based approaches (Haaswijk et al., 2018; Hosny et al., 2020; Zhu et al., 2020). Neto et al. (2019) partition the original graph into smaller sub-networks and performs binary classification on sub-networks to pick which recipes work best. On the other hand, RL-based solutions Haaswijk et al. (2018); Hosny et al. (2020); Zhu et al. (2020) use online RL algorithms to craft synthesis recipes, but do not leverage prior data. We show that ABC-RL outperforms them.

**ML for EDA:** ML has been used for a range of EDA problems Mirhoseini et al. (2021); Kurin et al. (2020); Lai et al. (2022; 2023); Schmitt et al. (2021); Yolcu & Póczos (2019); Vasudevan et al. (2021); Yang et al. (2022). Closer to this work, Mirhoseini et al. (2021) used a deep-RL agent to optimize chip placement, a different problem, and use the pre-trained agent (with online fine-tuning) to place the new design. This leaves limited scope for online exploration. Additionally, each move or action in placement, i.e., moving the x-y co-ordinates of modules in the design, is cheap unlike time-consuming actions in logic synthesis. Thus placement agents can be fine-tuned with larger amounts of test-time data relative to ABC-RL which has a constrained online search budget. Our ablation study shows ABC-RL defeats search combined with fine-tuned agent for given synthesis budget. A related body of work developed general representations of boolean circuits, for instance, DeepGate Li et al. (2022); Shi et al. (2023), ConVERTS Chowdhury et al. (2023) and "functionality matters" Wang et al. (2022), learned on signal probability estimation and functionality prediction, respectively. These embeddings could enhance the quality of our GCN embeddings and are interesting avenues for future work.

**RL and search for combinatorial optimization:** Fusing learning and search finds applications across diverse domains such as branching heuristics (He et al., 2014), Go and chess playing (Silver et al., 2016; Schrittwieser et al., 2020), traveling salesman (TSP) (Xing & Tu, 2020), and common subgraph detection (Bai et al., 2021). Each of these problems has unique structure. TSP and common subgraph detection both have graph inputs like logic synthesis but do not perform transformations on graphs. Branching problems have tree-structure, but do not operate on graphs. Go and Chess involve self-play during training and must anticipate opponents. Thus these works have each developed specialized solutions tailored to the problem domain, as we do with ABC-RL. Further, these previous works have not identified distribution shift as a problem and operate atleast under the assumption that train-test state distributions align closely.

**Retrieval guided Reinforcement learning:** Recent works (Goyal et al., 2022; Humphreys et al., 2022) have explored the benefits of retrieval in game-playing RL-agents. However, they implement retrieval differently: trajectories from prior episodes are retrieved and the entire trajectory is an *additonal* input to the policy agent. This also requires the policy-agent to be aware of retrieval during training. In contrast, our retrieval strategy is *lightweight*; instead of an entire graph/netlist, we only retrieve the similarity score from the training dataset and then fix $\alpha$. In addition, we do not need to incorporate the retrieval strategy during training, enabling off-the-shelf use of pre-trained RL agents. ABC-RL already significantly outperforms SOTA methods with this strategy, but the approach might be beneficial in other settings where online costs are severely constrained.

## 5 CONCLUSION

We introduce ABC-RL, a novel methodology that optimizes learning and search through a retrieval-guided mechanism, significantly enhancing the identification of high-quality synthesis recipes for new hardware designs. Specifically, tuning the $\alpha$ parameter of the RL agent during MCTS search within the synthesis recipe space effectively mitigates misguided searches toward unfavorable rewarding trajectories, particularly when encountering sufficiently novel designs. These core concepts, substantiated by empirical results, underscore the potential of ABC-RL in generating high-quality synthesis recipes, thereby streamlining modern complex chip design processes for enhanced efficiency.

**Reproducibility Statement** For reproducibility, we provide detailed information regarding methodologies, architectures, and settings in Section 3.1.

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

# A    APPENDIX 1

## A.1    LOGIC SYNTHESIS

Logic synthesis transforms a hardware design in register transfer level (RTL) to a Boolean gate-level network, optimizes the number of gates/depth, and then maps it to standard cells in a technology library Brayton et al. (1984). Well-known representations of Boolean networks include sum-of-product form, product-of-sum, Binary decision diagrams, and AIGs which are a widely accepted format using only AND (nodes) and NOT gates (dotted edges). Several logic minimization heuristics (discussed in Section A.2)) have been developed to perform optimization on AIG graphs because of its compact circuit representation and directed acyclic graph (DAG)-based structuring. These heuristics are applied sequentially ("synthesis recipe") to perform one-pass logic optimization reducing the number of nodes and depth of AIG. The optimized network is then mapped using cells from technology library to finally report area, delay and power consumption.

## A.2    LOGIC MINIMIZATION HEURISTICS

We now describe optimization heuristics provided by industrial strength academic tool ABC Brayton & Mishchenko (2010):

**1. Balance (b)** optimizes AIG depth by applying associative and commutative logic function tree-balancing transformations to optimize for delay.

**2. Rewrite (rw, rw -z)** is a directed acyclic graph (DAG)-aware logic rewriting technique that performs template pattern matching on sub-trees and encodes them with equivalent logic functions.

**3. Refactor (rf, rf -z)** performs aggressive changes to the netlist without caring about logic sharing. It iteratively examines all nodes in the AIG, lists out the maximum fan-out-free cones, and replaces them with equivalent functions when it improves the cost (e.g., reduces the number of nodes).

**4. Re-substitution (rs, rs -z)** creates new nodes in the circuit representing intermediate functionalities using existing nodes; and remove redundant nodes. Re-substitution improves logic sharing.

The zero-cost (-z) variants of these transformation heuristics perform structural changes to the netlist without reducing nodes or depth of AIG. However, previous empirical results show circuit transformations help future passes of other logic minimization heuristics reduce the nodes/depth and achieve the minimization objective.

## A.3    MONTE CARLO TREE SEARCH

We discuss in detail the MCTS algorithm. During selection, a search tree is built from the current state by following a search policy, with the aim of identifying promising states for exploration.

where $Q_{MCTS}^k(s, a)$ denotes estimated $Q$ value (discussed next) obtained after taking action $a$ from state $s$ during the $k^{th}$ iteration of MCTS simulation. $U_{MCTS}^k(s, a)$ represents upper confidence tree (UCT) exploration factor of MCTS search.

$$U_{MCTS}^k(s, a) = c_{\text{UCT}} \sqrt{\frac{\log\left(\sum_a N_{MCTS}^k(s, a)\right)}{N_{MCTS}^k(s, a)}}, \tag{5}$$

$N_{MCTS}^k(s, a)$ denotes the visit count of the resulting state after taking action $a$ from state $s$. $c_{UCT}$ denotes a constant exploration factor Kocsis & Szepesvári (2006).

The selection phase repeats until a leaf node is reached in the search tree. A leaf node in MCTS tree denotes either no child nodes have been created or it is a terminal state of the environment. Once a leaf node is reached the expansion phase begins where an action is picked randomly and its roll out value is returned or $R(s_L)$ is returned for the terminal state $s_L$. Next, back propagation happens where all parent nodes $Q_k(s, a)$ values are updated according to the following equation.

$$Q_{MCTS}^k(s, a) = \sum_{i=1}^{N_{MCTS}^k(s, a)} R_{MCTS}^i(s, a) / N_{MCTS}^k(s, a). \tag{6}$$

## A.4 ABC-RL AGENT PRE-TRAINING PROCESS

As discussed in Section 2.3, we pre-train an agent using available past data to help with choosing which logic minimization heuristic to add to the synthesis recipe. The process is shown as Algorithm 1.

---

**Algorithm 1** ABC-RL: Policy agent pre-training

---

1: **procedure** TRAINING($\theta$)
2:     Replay buffer $(RB) \leftarrow \phi$, $\mathcal{D}_{train} = \{AIG_1, AIG_2, ..., AIG_n\}$, num_epochs=$N$, Recipe length=$L$, AIG embedding network: $\Lambda$, Recipe embedding network: $\mathcal{R}$, Agent policy $\pi_\theta := U$ (Uniform distribution), MCTS iterations = $K$, Action space = $A$
3:     **for** $AIG_i \in \mathcal{D}_{train}$ **do**
4:         $r \leftarrow 0, depth \leftarrow 0$
5:         $s \leftarrow \Lambda(AIG_i) + \mathcal{R}(r)$
6:         **while** depth < $L$ **do**
7:             $\pi_{MCTS} = MCTS(s, \pi_\theta, K)$
8:             $a = argmax_{a' \in A} \pi_{MCTS}(s, a')$
9:             $r \leftarrow r + a, s' \leftarrow \mathcal{A}(AIG_i) + \mathcal{R}(r)$
10:           $RB \leftarrow RB \bigcup (s, a, s', \pi_{MCTS}(s, \cdot))$
11:           $s \leftarrow s', depth \leftarrow depth + 1$
12:     **for** epochs < $N$ **do**
13:         $\theta \leftarrow \theta_i - \alpha \nabla_\theta \mathcal{L}(\pi_{MCTS}, \pi_\theta)$

---

## B NETWORK ARCHITECTURE

### B.1 AIG NETWORK ARCHITECTURE

**AIG encoding in ABC**: An AIG graph is a directed acyclic graph representing the circuit's boolean functionality. We read in the same AIG format introduced in Mishchenko et al. (2006) and commonly used in literature: nodes in the AIG represent AND gates, Primary Inputs (PIs) or Primary Outputs (POs). On the other hand, NOT gates are represented by edges: dashed edges represent NOT gates (i.e., the output of the edge is a logical negation of its input) and solid edges represent a simple wire whose output equals its input.

**GCN-based AIG embedding**: Starting with a graph G = ($\mathbf{V}$, $\mathbf{E}$) that has vertices $\mathbf{V}$ and edges $\mathbf{E}$, the GCN aggregates feature information of a node with its neighbors' node information. The output is then normalized using `Batchnorm` and passed through a non-linear `LeakyReLU` activation function. This process is repeated for k layers to obtain information for each node based on information from its neighbours up to a distance of k-hops. A graph-level READOUT operation produces a graph-level embedding. Formally:

$$h_u^k = \sigma(W_k \sum_{i \in u \cup N(u)} \frac{h_i^{k-1}}{\sqrt{N(u)} \times \sqrt{N(v)}} + b_k), k \in [1..K] \tag{7}$$

$$h_G = READOUT(\{h_u^k; u \in V\})$$

Here, the embedding for node $u$, generated by the $k^{th}$ layer of the GCN, is represented by $h_u^k$. The parameters $W_k$ and $b_k$ are trainable, and $\sigma$ is a non-linear ReLU activation function. $N(\cdot)$ denotes the 1-hop neighbors of a node. The READOUT function combines the activations from the $k^{th}$ layer of all nodes to produce the final output by performing a pooling operation.

Each node in the AIG read in from ABC is translated to a node in our GCN. For the initial embeddings, $h_u^0$, We use two-dimensional vector to encode node-level features: (1) node type (AND, PI, or PO) and (2) number of negated fan-in edges Chowdhury et al. (2021; 2022). we choose $k = 3$ and global average and max pooling concatenated as the READOUT operation.

**Architectural choice of GNN**: We articulate our rationale for utilizing a simple Graph Convolutional Network (GCN) architecture to encode AIGs for the generation of synthesis recipes aimed at optimizing the area-delay product. We elucidate why this approach is effective and support our argument with an experiment that validates its efficacy:

- **Working principle of logic synthesis transformations**: Logic synthesis transformations of ABC (and in general commercial logic synthesis tools) including *rewrite*, *refactor* and *re-substitute* performs transformations at a local subgraph levels rather than the whole AIG structure. For e.g. *rewrite* performs a backward pass from primary outputs to primary inputs, perform k-way cut at each AIG node and replace the functionality with optimized implementation available in the truth table library of ABC. Similarly, *refactor* randomly picks a fan-in cone of an intermediate node of AIG and replaces it with a different implementation if it reduced the nodes of AIG. Thus, effectiveness of any synthesis transformations do not require deeper GCN layers; capturing neighborhood information upto depth 3 in our case worked well to extract features out of AIG which can help predict which transformation next can help reduce area-delay product.

- **Feature initialization of nodes in AIG**: The node in our AIG encompasses two important feature: i) Type of node (Primary Input, Primary Output and Internal Node) and ii) Number of negated fan-ins. Thus, our feature initialization scheme takes care of the functionality even though the structure of AIG are exactly similar and the functionality. Thus, two AIG having exact same structure but edge types are different (dotted edge represent negation and solid edge represent buffer), the initial node features of AIG itself will be vastly different and thus our 3-layer GCNs have been capable enough to distinguish between them and generate different synthesis recipes.

Several recent GNN-based architectures Li et al. (2022); Shi et al. (2023) have been proposed recently to capture functionality of AIG-based hardware representations. This remains an active exploration direction to further enhance the benefits of ABC-RL to distinguish structurally similar yet substantially varying in functionality space.

## C  EXPERIMENTAL DETAILS

### C.1  REWARD NORMALIZATION

In our work, maximizing QoR entails finding a recipe $P$ which is minimizing the area-delay product of transformed AIG graph. We consider as a baseline recipe an expert-crafted synthesis recipe `resyn2` Mishchenko et al. (2006) on top of which we improve our ADP.

$$R = \begin{cases} 1 - \frac{ADP(\mathcal{S}(G,P))}{ADP(\mathcal{S}(G,resyn2))} & ADP(\mathcal{S}(G,P)) < 2 \times ADP(\mathcal{S}(G,P)), \\ -1 & otherwise. \end{cases}$$

### C.2  BENCHMARK CHARACTERIZATION

We present the characterization of circuits used in our dataset. This data provides a clean picture on size and level variation across all the AIGs.

| Name | Inputs | Outputs | Nodes | level |
|---|---|---|---|---|
| alu2 | 10 | 6 | 401 | 40 |
| alu4 | 10 | 6 | 735 | 42 |
| apex1 | 45 | 45 | 2655 | 27 |
| apex2 | 39 | 3 | 445 | 29 |
| apex3 | 54 | 50 | 2374 | 21 |
| apex4 | 9 | 19 | 3452 | 21 |
| apex5 | 117 | 88 | 1280 | 21 |
| apex6 | 135 | 99 | 659 | 15 |
| apex7 | 49 | 37 | 221 | 14 |
| b2 | 16 | 17 | 1814 | 22 |
| b9 | 41 | 21 | 105 | 10 |
| C432 | 36 | 7 | 209 | 42 |
| C499 | 41 | 32 | 400 | 20 |
| C880 | 60 | 26 | 327 | 24 |
| C1355 | 41 | 32 | 504 | 26 |
| C1908 | 31 | 25 | 414 | 32 |
| C2670 | 233 | 140 | 717 | 21 |
| C3540 | 50 | 22 | 1038 | 41 |
| C5315 | 178 | 123 | 1773 | 38 |
| C6288 | 32 | 32 | 2337 | 120 |
| C7552 | 207 | 108 | 2074 | 29 |
| frg1 | 28 | 3 | 126 | 19 |
| frg2 | 143 | 139 | 1164 | 13 |
| i10 | 257 | 224 | 2675 | 50 |
| i7 | 199 | 67 | 904 | 6 |
| i8 | 133 | 81 | 3310 | 21 |
| i9 | 88 | 63 | 889 | 14 |
| m3 | 8 | 16 | 434 | 14 |
| m4 | 8 | 16 | 760 | 14 |
| max1024 | 10 | 6 | 1021 | 20 |
| max128 | 7 | 24 | 536 | 13 |
| max512 | 9 | 6 | 743 | 19 |
| pair | 173 | 137 | 1500 | 24 |
| prom1 | 9 | 40 | 7803 | 24 |
| prom2 | 9 | 21 | 3513 | 22 |
| seq | 41 | 35 | 2411 | 29 |
| table3 | 14 | 14 | 2183 | 24 |
| table5 | 17 | 15 | 1987 | 26 |
| adder | 256 | 129 | 1020 | 255 |
| bar | 135 | 128 | 3336 | 12 |
| div | 128 | 128 | 44762 | 4470 |
| log2 | 32 | 32 | 32060 | 444 |
| max | 512 | 130 | 2865 | 287 |
| multiplier | 128 | 128 | 27062 | 274 |
| sin | 24 | 25 | 5416 | 225 |
| sqrt | 128 | 64 | 24618 | 5058 |
| square | 62 | 128 | 18484 | 250 |
| arbiter | 256 | 129 | 11839 | 87 |
| ctrl | 7 | 26 | 174 | 10 |
| cavlc | 10 | 11 | 693 | 16 |
| i2c | 147 | 142 | 1342 | 20 |
| int2float | 11 | 7 | 260 | 16 |
| mem_ctrl | 1204 | 1231 | 46836 | 114 |
| priority | 128 | 8 | 978 | 250 |
| router | 60 | 30 | 257 | 54 |
| voter | 1001 | 1 | 13758 | 70 |

**Table 5:** Benchmark characterization: Primary inputs, outputs, number of nodes and level of AIGs

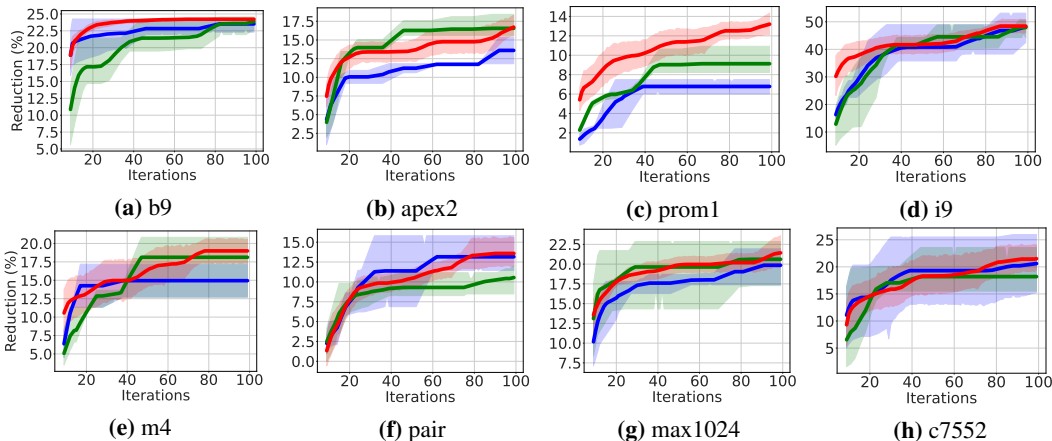

**Figure 7:** Area-delay product reduction (in %) compared to resyn2 on MCNC circuits. GREEN: SA+Pred. Chowdhury et al. (2022), BLUE: MCTS Neto et al. (2022), RED: ABC-RL

# D RESULTS

## D.1 PERFORMANCE OF ABC-RL AGAINST PRIOR WORKS AND BASELINE MCTS+LEARNING

### D.1.1 MCNC BENCHMARKS

Figure 7 plots the ADP reductions over search iterations for MCTS, SA+Pred, and ABC-RL. In `m4`, ABC-RL's agent explores paths with higher rewards whereas standard MCTS continues searching without further improvement. A similar trend is observed for `prom1` demonstrating that a pre-trained agent helps bias search towards better parts of the search space. SA+Pred. Chowdhury et al. (2022) also leverages past history, but is unable to compete (on average) with MCTS and ABC-RL in part because SA typically underperforms MCTS on tree-based search spaces. Also note from Figure 5 that ABC-RL in most cases achieves higher ADP reductions earlier than competing methods (except `pair`). This results in significant geo. mean run-time speedups of $2.5\times$ at iso-QoR compared to standard MCTS on MCNC benchmarks.

### D.1.2 EPFL ARITHMETIC BENCHMARKS

Figure 8 illustrates the performance of ABC-RL in comparison to state-of-the-art methods: Pure MCTS Neto et al. (2022) and SA+Prediction Chowdhury et al. (2022). In contrast to the scenario where MCTS+Baseline underperforms pure MCTS (as shown in 2), here we observe that ABC-RL effectively addresses this issue, resulting in superior ADP reduction. Remarkably, ABC-RL achieved a geometric mean $5.8\times$ iso-QoR speed-up compared to MCTS across the EPFL arithmetic benchmarks.

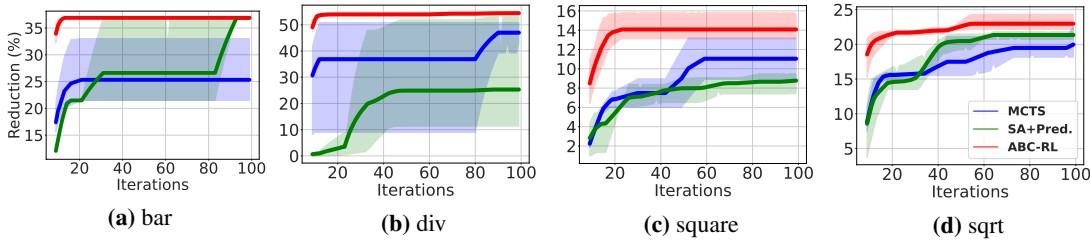

**Figure 8:** Area-delay product reduction (in %) compared to resyn2 on EPFL arithmetic benchmarks. GREEN: SA+Pred. Chowdhury et al. (2022), BLUE: MCTS Neto et al. (2022), RED: ABC-RL

### D.1.3 EPFL RANDOM CONTROL BENCHMARKS

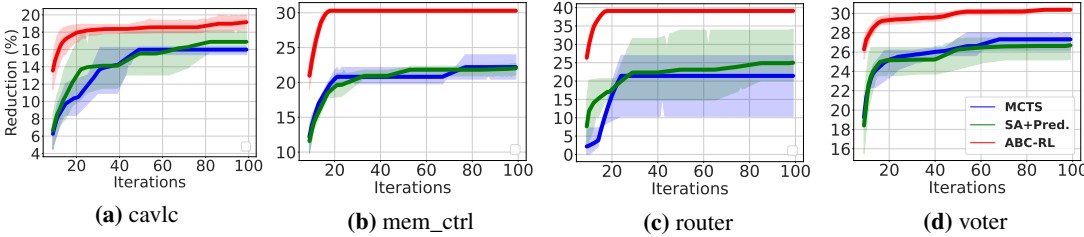

| (a) cavlc | (b) mem_ctrl | (c) router | (d) voter |

**Figure 9:** Area-delay product reduction (in %) compared to resyn2 on EPFL random control benchmarks. On `cavlc` and `router`, ABC-RL perform better than MCTS where baseline MCTS+Learning under-perform. GREEN: SA+Pred. Chowdhury et al. (2022), BLUE: MCTS Neto et al. (2022), RED: ABC-RL.

### D.2 PERFORMANCE OF BENCHMARK-SPECIFIC ABC-RL AGENTS

**ABC-RL+MCNC agent:** For 6 out of 12 MCNC benchmarks, ABC-RL guided by the MCNC agent demonstrated improved performance compared to the benchmark-wide agent. This suggests that the hyper-parameters ($\delta_{th}$ and $T$) derived from the validation dataset led to optimized $\alpha$ values for MCNC benchmarks. However, the performance of the MCNC agent was comparatively lower on EPFL arithmetic and random control benchmarks.

**ABC-RL+ARITH agent:** Our EPFL arith agent resulted in better ADP reduction compared to benchmark-wide agent only on A4(`sqrt`). This indicate that benchmark-wide agent is able to learn more from diverse set of benchmarks resulting in better ADP reduction. On MCNC benchmarks, we observe that ARITH agent performed the best amongst all on C6(`m4`) and C10 (`c7552`) because these are arithmetic circuits.

**ABC-RL+RC agent:** Our RC agent performance on EPFL random control benchmarks are not that great compared to benchmark-wide agent. This is primarily because of the fact that EPFL random control benchmarks have hardware designs performing unique functionality and hence learning from history doesn't help much. But, ABC-RL ensures that performance don't detoriate compared to pure MCTS.

### D.3 PERFORMANCE OF ABC-RL VERSUS FINE-TUNING (MCTS+L+FT)

**MCNC Benchmarks:** In Fig. 10, we depict the performance comparison among MCTS+finetune agent, ABC-RL, and pure MCTS. Remarkably, ABC-RL outperforms MCTS+finetune on 11 out of 12 benchmarks, approaching MCTS+finetune's performance on `b9`. A detailed analysis of circuits where MCTS+finetune performs worse than pure MCTS (`i9`, `m4`, `pair`, `c880`, `max1024`, and `c7552`) reveals that these belong to 6 out of 8 MCNC designs where MCTS+learning performs suboptimally compared to pure MCTS. This observation underscores the fact that although finetuning contributes to a better geometric mean over MCTS+learning (23.3% over 20.7%), it still falls short on 6 out of 8 benchmarks. For the remaining two benchmarks, `alu4` and `apex4`, MCTS+finetune performs comparably to pure MCTS for `alu4` and slightly better for `apex4`. Thus, ABC-RL emerges as a more suitable choice for scenarios where fine-tuning is resource-intensive, yet we seek a versatile agent capable of appropriately guiding the search away from unfavorable trajectories.

**EPFL Benchmarks:** In Fig. 11 and 12, we present the performance comparison with MCTS+finetune. Notably, for designs `bar` and `div`, MCTS+finetune achieved equivalent ADP as ABC-RL, maintaining the same iso-QoR speed-up compared to MCTS. These designs exhibited strong performance with baseline MCTS+Learning, thus aligning with the expectation of favorable results with MCTS+finetune. On `square`, MCTS+finetune nearly matched the ADP reduction achieved by pure MCTS. This suggests that fine-tuning contributes to policy improvement from the pre-trained agent, resulting in enhanced performance compared to baseline MCTS+Learning. In the case of `sqrt`, MCTS+finetune approached the performance of ABC-RL. Our fine-tuning experiments affirm its ability to correct the model policy, although it require more samples to converge towards ABC-RL performance.

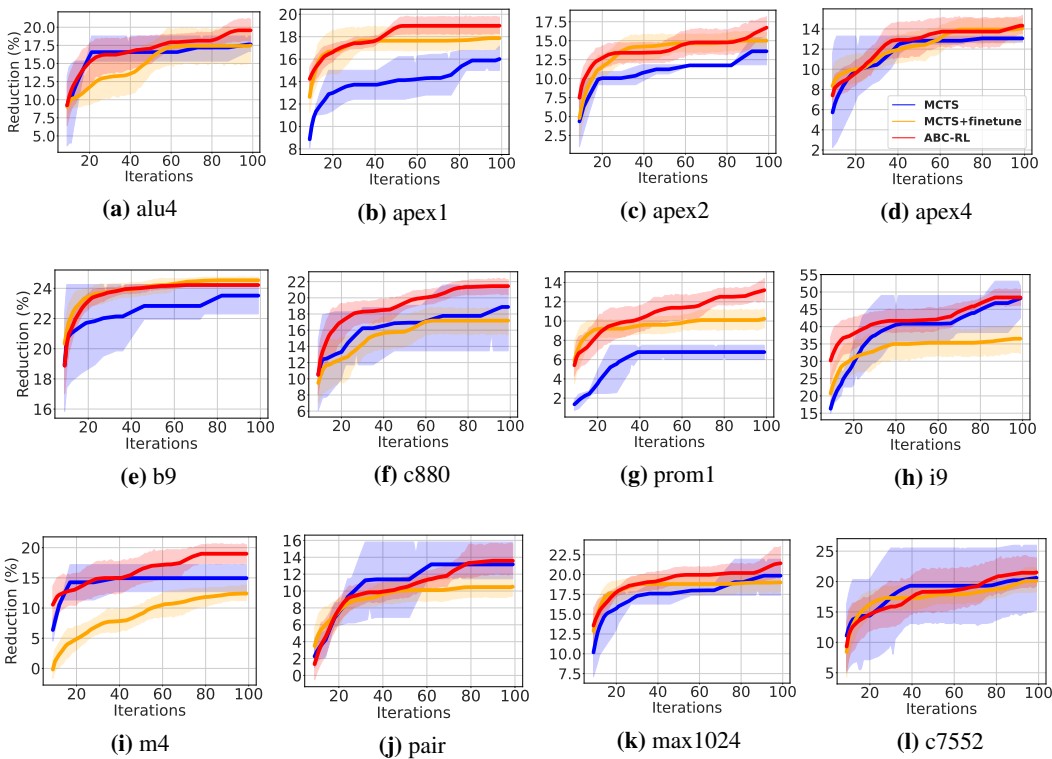

**Figure 10:** Area-delay product reduction (in %) compared to resyn2 on MCNC benchmarks. YELLOW: MCTS+Finetune, BLUE: MCTS Neto et al. (2022), RED: ABC-RL

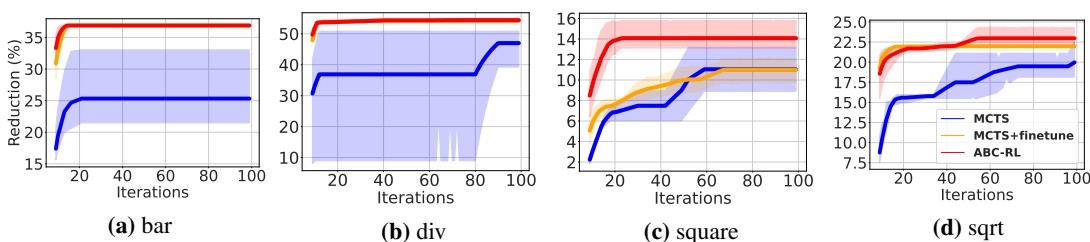

**Figure 11:** Area-delay product reduction (in %) compared to resyn2 on EPFL arithmetic benchmarks. YELLOW: MCTS+Finetune, BLUE: MCTS Neto et al. (2022), RED: ABC-RL

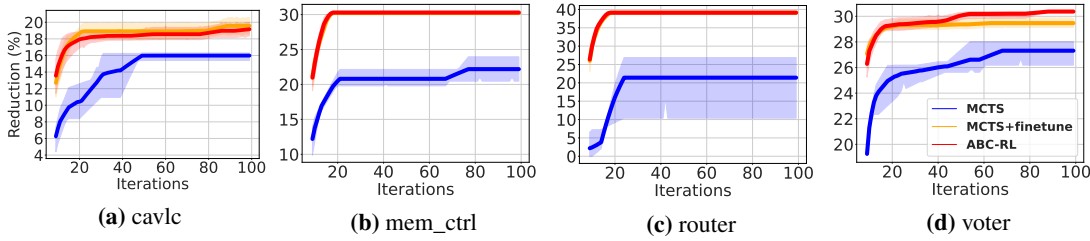

**Figure 12:** Area-delay product reduction (in %) compared to resyn2 on EPFL random control benchmarks. YELLOW: MCTS+FT, BLUE: MCTS Neto et al. (2022), RED: ABC-RL

## D.4 ABC-RL: SIMILARITY SCORE COMPUTATION AND NEAREST NEIGHBOUR RETRIEVAL

Next, we report nearest neighbor retrieval performance of ABC-RL which is a key mechanism in setting $\alpha$ to tune pre-trained agent's recommendation during MCTS search. We report similarity score which is the cosine distance between test AIG and nearest neighbour retrieved via similarity. We also report the training circuit which the test AIG is closest to. Based on our validation dataset, we set $T = 100$ and $\delta_{th} = 0.007$.

| Benchmark | Designs | Similarity Score | Nearest neighbour |
|---|---|---|---|
| MCNC | alu4 | 0.000 | alu2 |
| | apex1 | 0.001 | apex3 |
| | apex2 | 0.002 | alu2 |
| | apex4 | 0.000 | prom2 |
| | c7552 | 0.006 | max512 |
| | i9 | 0.003 | prom2 |
| | m4 | 0.001 | max512 |
| | prom1 | 0.002 | prom2 |
| | b9 | 0.005 | c2670 |
| | c880 | 0.006 | frg1 |
| | pair | 0.010 | m3 |
| | max1024 | 0.023 | alu2 |
| EPFL arith | bar | 0.002 | prom2 |
| | div | 0.001 | log2 |
| | square | 0.012 | alu2 |
| | sqrt | 0.002 | multiplier |
| EPFL random control | cavlc | 0.007 | alu2 |
| | mem_ctrl | 0.001 | prom2 |
| | router | 0.025 | adder |
| | voter | 0.006 | m3 |

**Table 6:** Similarity score ($\times 10^{-2}$) of nearest neighbour retrieved using ABC-RL for test designs. Nearest neighbour denotes the training design closest to test-time design

## D.5 ABC-RL: ARCHITECTURAL CHOICES FOR RECIPE ENCODER

ABC-RL uses BERT-based recipe encoder to extract two meaningful information: 1) Contextual relationship between current synthesis transformations and previous ones and 2) The synthesis transformations which needs more attention compared to others depending on its position. For e.g. `rewrite` operation at the start of a synthesis recipe tend to optimize more number of nodes in AIG compared to its position later in the recipe Yu (2020); Neto et al. (2022). Similarly, transformations like `balance` are intended towards reducing delay of the design wheras transformations like `rewrite, refactor, resub` are intended towards area optimization. Thus, selective attention and placement of their positions with respect to other synthesis transformations needs to be learned which makes BERT an ideal choice to encode synthesis recipe. As part of additional ablation study, we encode our synthesis recipe with LSTM network with input sequence length ($L = 10$) and apply zero-padding for recipe length less than $L$.

**Table 7:** Area-delay reduction (in %). ABC-RL$-$BERT is ABC-RL trained with naive synthesis encoder instead of BERT. MCTS+L+FT indicate MCTS+Learning with online fine-tuning.

| Recipe encoder | ADP reduction (in %) | | | | | | | | | | | | | | | | | | | | Geo. mean |
|---|---|---|---|---|---|---|---|---|---|---|---|---|---|---|---|---|---|---|---|---|---|
| | MCNC | | | | | | | | | | | | EPFL arith | | | | EPFL random | | | | |
| | C1 | C2 | C3 | C4 | C5 | C6 | C7 | C8 | C9 | C10 | C11 | C12 | A1 | A2 | A3 | A4 | R1 | R2 | R3 | R4 | |
| Naive | 17.0 | 16.5 | 14.9 | 13.1 | 44.6 | 16.9 | 10.0 | 23.5 | 16.3 | 18.8 | 10.6 | 19.0 | 36.9 | 51.7 | 9.9 | 20.0 | 15.5 | 23.0 | 28.0 | 26.9 | 21.2 |
| LSTM-based | 19.5 | 17.8 | 14.9 | 13.3 | 44.5 | 17.8 | 10.3 | 23.5 | 18.8 | 20.1 | 10.6 | 19.6 | 36.9 | 53.4 | 10.9 | 22.4 | 16.8 | 24.3 | 32.3 | 28.7 | 22.6 |
| BERT-based | 19.9 | 19.6 | 16.8 | 15.0 | 46.9 | 19.1 | 12.1 | 24.3 | 21.3 | 21.1 | 13.6 | 21.6 | 36.9 | 56.2 | 14.0 | 23.8 | 19.8 | 30.2 | 38.9 | 30.0 | 25.3 |

## D.6   RUNTIME ANALYSIS OF ABC-RL VERSUS SOTA

We now present wall-time comparison of ABC-RL versus SOTA methods on test designs for 100 iterations. We note that for all online search schemes, **the runtime is dominated by that the number of online synthesis runs (typically 9.5 seconds per run) as opposed to inference cost of the deep network (*e.g.* 11 milli-seconds for ABC-RL)** . Thus, as observed in the table below, ABC-RL runtime is within 1.5% of MCTS and SA+Pred. Table 8 presents wall-time comparison of ABC-RL versus existing SOTA methods for 100 iterations. Overall, ABC-RL run time overhead over MCTS and SA+Pred. (over 100 iterations) has a geometric mean of 1.51% and 2.09%, respectively. In terms of wall-time, ABC-RL achieves 3.75x geo. mean iso-QoR speed-up.

| Designs | Runtime (in seconds) | | | ABC-RL overhead (in %) | | Iso-QoR speed-up |
| | MCTS | SA+Pred. | ABC-RL | w.r.t. MCTS | w.r.t. SA+Pred. | w.r.t. MCTS wall-time |
| --- | --- | --- | --- | --- | --- | --- |
| alu4 | 35.6 | 35.3 | 36.0 | 1.12 | 1.98 | 1.89x |
| apex1 | 105.6 | 105.1 | 107.0 | 1.33 | 1.81 | 5.82x |
| apex2 | 20.2 | 20.1 | 20.5 | 1.49 | 1.99 | 1.80x |
| apex4 | 195.6 | 193.7 | 199.6 | 2.04 | 3.05 | 1.58x |
| c7552 | 91.4 | 91.2 | 93.2 | 1.97 | 2.19 | 1.16x |
| i9 | 40.2 | 40.3 | 41.2 | 2.49 | 2.23 | 1.30x |
| m4 | 37.1 | 37.3 | 37.9 | 1.89 | 1.41 | 3.11x |
| prom1 | 201.6 | 200.3 | 205.7 | 2.03 | 2.70 | 4.04x |
| b9 | 16.9 | 16.8 | 17.3 | 2.37 | 2.98 | 2.11x |
| c880 | 16.8 | 16.6 | 17.5 | 2.38 | 2.99 | 1.20x |
| pair | 110.3 | 110.0 | 112.0 | 1.54 | 1.73 | 0.90x |
| max1024 | 82.6 | 82.0 | 84.9 | 2.78 | 3.50 | 1.77x |
| bar | 192.4 | 191.2 | 196.5 | 2.13 | 2.77 | 3.59x |
| div | 655.4 | 652.1 | 668.9 | 2.06 | 2.58 | 8.82x |
| square | 398.6 | 395.0 | 403.1 | 1.12 | 2.05 | 4.17x |
| sqrt | 448.5 | 444.2 | 455.5 | 1.56 | 2.54 | 8.21x |
| cavlc | 56.4 | 56.1 | 57.2 | 1.42 | 1.96 | 4.95x |
| mem_ctrl | 953.7 | 950.3 | 966.5 | 1.34 | 1.70 | 6.33x |
| router | 44.2 | 43.9 | 44.8 | 1.36 | 2.05 | 3.08x |
| voter | 312.5 | 311.0 | 317.9 | 1.73 | 2.22 | 5.57x |
| Geomean | - | - | - | 1.51 | 2.09 | 3.75x |

**Table 8:** Wall-time overhead of ABC-RL over SOTA methods for 100 iterations (Budget: 100 synthesis runs). We report iso-QoR wall-time speed-up with respect to baseline MCTS Neto et al. (2022).

## D.7   ABC-RL PERFORMANCE ON TRAINING AND VALIDATION DESIGNS

Next, we present ABC-RL performance on training and validation circuits and compare it with baseline MCTS. For training circuits, ABC-RL sets $\alpha = 0$ indicating the search with augmented with full recommendation from pre-trained agent. For validation circuits, ABC-RL sets $\alpha$ and performs search with tuned $\alpha$ recommendation from pre-trained agent.

| Designs | ADP reduction (in %) | |
| --- | --- | --- |
| | MCTS | ABC-RL |
| alu2 | 21.2 | 22.3 |
| apex3 | 12.9 | 12.9 |
| apex5 | 32.50 | 32.5 |
| apex6 | 10.00 | 10.7 |
| apex7 | 0.80 | 1.7 |
| b2 | 20.75 | 22.1 |
| c1355 | 34.80 | 35.4 |
| c1908 | 14.05 | 15.9 |
| c2670 | 7.50 | 10.7 |
| c3540 | 20.30 | 22.8 |
| c432 | 31.00 | 31.8 |
| c499 | 12.80 | 12.8 |
| c6288 | 0.28 | 0.6 |
| frg1 | 25.80 | 25.8 |
| frg2 | 46.20 | 47.1 |
| i10 | 28.25 | 31.2 |
| i7 | 37.50 | 37.7 |
| i8 | 40.00 | 47.3 |
| m3 | 20.10 | 25.0 |
| max128 | 24.10 | 31.8 |
| max512 | 14.00 | 16.8 |
| prom2 | 18.10 | 19.8 |
| seq | 17.10 | 20.9 |
| table3 | 15.95 | 16.1 |
| table5 | 23.40 | 25.5 |
| Geomean | 15.39 | 17.84 |

**Table 9:** Area-delay reduction compared to `resyn2` on MCNC training and validation circuits. We compare results of MCTS and ABC-RL approach.

| Designs | ADP reduction (in %) | | Designs | ADP reduction (in %) | |
| --- | --- | --- | --- | --- | --- |
| | MCTS | ABC-RL | | MCTS | ABC-RL |
| adder | 18.63 | 18.63 | arbiter | 0.03 | 0.03 |
| log2 | 9.09 | 11.51 | ctrl | 27.58 | 30.85 |
| max | 37.50 | 45.86 | i2c | 13.45 | 15.65 |
| multiplier | 9.90 | 12.68 | int2float | 8.10 | 8.1 |
| sin | 14.50 | 15.96 | priority | 77.53 | 77.5 |
| Geomean | 15.55 | 18.18 | Geomean | 5.87 | 6.19 |

**Table 10:** Area-delay reduction over `resyn2` on EPFL arithmetic (left) and random control (right) training and validation benchmarks using MCTS and ABC-RL

