# OpenReview forum: "Retrieval-Guided Reinforcement Learning for Boolean Circuit Minimization"
_ICLR.cc/2024/Conference — ICLR 2024 poster_

### Official Review · Reviewer_pgL7 · 2023-10-23

**Soundness:** 4 excellent
**Presentation:** 4 excellent
**Contribution:** 3 good
**Rating:** 8
**Confidence:** 4

**Summary:**

The authors proposed a new method for finding optimal synthesis recipes for unseen netlists. They proposed three methods, one based on MCTS and a trained policy on training netlist (MCTS+L) and the other, and more superior, is using MCTS and a trained policy with a variable to control the reliance on the trained policy or the online MCTS search (ABC-RL). The variable \alpha is determined based on a nearest neighbor search to determine if the netlist is close to the training netlists or not. The authors split the dataset of netlists into train, validation, and test sets and used the train to train the policy and used the validation set to set the hyperparameter that controls \alpha. The experimental results show that the ABC-RL can outperform the other methods and SOTA in most of the test netlists and can achieve the best goe-mean performance.

**Strengths:**

- Extensive experimental results.
  - The authors provided comparisons with multiple baseline including Online-RL, Simulated Annealing, MCTS, and MCTS+L and MCTS+L+FT.
  - The study on training the policy on specific benchmarks was helpful showing the effect of \alpha and closeness of netlist to the training dataset.
  - They perform an ablation study on the architecture of the policy to determine the impact of the transformer architecture on the performance.
- The authors motivated the need for a variable that can control the effect of policy vs MCTS very well by an example in the Introduction Section.
- The paper is well written and organized and can be followed easily by non-experts.
- The related literature has been sufficiently reviewed and cited.

**Weaknesses:**

- The idea of using \alpha was interesting and novel to the best of my knowledge, however it is a simple and small contribution.

**Questions:**

- It would be great if the authors can also provide the results for the rest of the benchmark netlists to ensure that the similar performance gains hold up for the training and validation sets.
- In Section 2.4, in definition of \sigma_{\delta_{th}, T}(z), replace \teta with \delta_{th}.

---

> ### Author Response · Authors · 2023-11-17
> **Official response to pgL7**
>
> We thank the reviewer for their detailed and insightful feedback. We address the questions raised:
>
> Q1) *It would be great if the authors can also provide the results for the rest of the benchmark netlists to ensure that the similar performance gains hold up for the training and validation sets.*
>
> **Response:**
>
> Thank you for your recommendation. To provide a comprehensive overview, we applied ABC-RL to both training and validation netlists, and the performance improvements are detailed in ***Appendix D7 (Table 9 and 10)***. On MCNC circuits, ABC-RL achieved a geometric mean ADP reduction of **17.84%**, surpassing MCTS, which resulted in a **15.39%** reduction. For EPFL arithmetic circuits, ABC-RL demonstrated an **18.18%** geometric mean reduction compared to MCTS, which yielded a **15.55%** ADP reduction. Finally, on EPFL random control circuits, the geometric mean ADP reduction was **6.19%** and **5.87%** for ABC-RL and MCTS, respectively.
>
>
> Q2) *In Section 2.4, in the definition of $\sigma_{\delta_{th}, T}(z)$, replace $\theta$ with $\delta_{th}$.*
>
> **Response:**
>
> Thanks for pointing out this oversight. We have replaced $\theta$ with $\delta_{th}$ in our updated manuscript.

---

### Official Review · Reviewer_yxFf · 2023-10-26

**Soundness:** 2 fair
**Presentation:** 3 good
**Contribution:** 2 fair
**Rating:** 6
**Confidence:** 5

**Summary:**

This paper proposes a new logical synthesis optimization sequence method ABC-RL based on MCTS and pertained policy. Unlike previous works, ABC-RL can compute the similarity of the new circuit with previous circuits, thus determining how much experience will be used. Experiments show that ABC-RL achieves SOTA optimization results in most circuits.

**Strengths:**

1. The idea that the degree of retrieval is determined by the similarity function is interesting, and it shows the advanced performance in experiments.

2. The experiments are extensive.

3. The paper is well-organized and easy to read.

**Weaknesses:**

1. Some other ML method baselines are not included. For example, the DRiLLS [1] results are not included in the paper, but it is an important method recently.

2. The retrieval performance is not well reported. For example, it should give us examples of each circuit’s nearest neighbor circuit and similarity factor.

3. ChiPFormer [2]  is an RL placement method with a pretrained policy and should be included in related work.

[1] Hosny, Abdelrahman, et al. "DRiLLS: Deep reinforcement learning for logic synthesis." 2020 25th Asia and South Pacific Design Automation Conference (ASP-DAC). IEEE, 2020.

[2] Lai, Yao, et al. "ChiPFormer: Transferable Chip Placement via Offline Decision Transformer." (2023).

**Questions:**

1. Could you compare ABC-RL and DRiLLS methods?

2. As weakness 2, could you give each circuit’s nest neighbor circuit and similarity factor?

---

> ### Author Response · Authors · 2023-11-17
> **Official response to yxFf**
>
> We thank the reviewer for their insightful feedback and address the concerns raised:
>
> Q1) *Some other ML method baselines are not included. For example, the DRiLLS [1] results are not included in the paper, but it is an important method recently.*
>
> **Response:**
>
> Thank you for the excellent suggestion. In the updated draft, we have cited and evaluated against DRiLLS as well, and reported results in ***Table 2***. DRiLLS is competitive with Online-RL (both have a **16.1%** ADP reduction), but has less ADP reduction compared to our proposed ABC-RL (**25.3%**).
>
> Q2) *The retrieval performance is not well reported. For example, it should give us examples of each circuit’s nearest neighbor circuit and similarity factor.*
>
> **Response:**
>
> Thank you for the suggestion. This is actually informative data and we have included it in ***Appendix D.4 (Table 6)***. Several interesting observations can be drawn: first, note that the closest circuits in the training set are often semantically similar (*alu2 for alu4, apex3 for apex1, sqrt for multiplier*), suggesting that the learned embeddings are capturing netlist structure/function. Second, netlists for which we found MCTS+L to underperform MCTS alone, for example, square and cavlc have relatively **high** cosine distances of 0.012 and 0.007, respectively, relative to the $\delta_{th}=0.007$ in Equation 4.
>
>
> Q3) *ChiPFormer [2] is an RL placement method with a pretrained policy and should be included in related work.*
>
> **Response:**
>
> Thank you for pointing out this recent work regarding pre-training offline agents for chip placement problems. We have included the citation in our related work section.
>
> [1] Hosny, Abdelrahman, et al. "DRiLLS: Deep reinforcement learning for logic synthesis." 2020 25th Asia and South Pacific Design Automation Conference (ASP-DAC). IEEE, 2020.
>
> [2] Lai, Yao, et al. "ChiPFormer: Transferable Chip Placement via Offline Decision Transformer." (2023).

---

> > ### Author Response · Authors · 2023-11-22
> >
> > We thank the reviewer again for their insightful questions. As the rebuttal period comes to a close, we are reaching out to highlight that we have directly addressed all questions that the reviewer asked and see if there are any remaining questions we could address. Specifically (i) as requested we also now compare with DRiLLs in the paper and show that ***ABC-RL outperforms DRiLLs***;  (ii) we have ***reported retrieval performance*** in Appendix D.4 (Table 6) and find that retrieved circuits are semantically close to the query circuits; and (iii) we have ***cited and discussed ChipFormer*** in the related work section.
> >
> > We would be glad to address any remaining questions the reviewer might have.

---

### Official Review · Reviewer_qBwB · 2023-10-30

**Soundness:** 2 fair
**Presentation:** 3 good
**Contribution:** 3 good
**Rating:** 6
**Confidence:** 4

**Summary:**

Summary:
 This papers proposes ABC-RL method to smoothly change the impact of learned policy in the search objective based on the graph similarity. Similar designs to training examples will have more bias from learned policy, while novel circuit will use more search. Compared to prior method, the propose one outperforms in various designs in terms of area-delay product.

**Strengths:**

Strength:

1.	The proposed method is based on good observation that the benchmark has a large diversity, and the learned policy sometimes is not helpful for novel circuits.

2.	The proposed method smoothly controls the importance of learned policy using GNN-based graph embedding similarity scores. The variable heuristic synthesis solution from the learned policy is encoded with transformer blocks for better performance.

3.	It also runs fast with runtime benefits.

**Weaknesses:**

Weakness:

1.	The overall framework basically does test-time augmentation. It assumes pre-trained policy is not generalizable to novel circuits, and by comparing the test example with the training example, it dynamically selects among two search strategies, but in a smooth way. The circuit similarity is a performance proxy for pretrained agent and MCTS method, and use that proxy to predict the weights to ensemble two models. The novelty, in this sense, is limited. Is it possible to combine more synthesis strategies based on a more general performance predictor at test time?

2.	The usage of BERT is not well justified. There are other simpler methods to encode variable-length sequences, e.g., RNN. The attention model is also data-hungry during training. Why BERT is the most suitable encoder?

3.	The assumption that learned policy cannot generalize to diverse benchmarks is not well supported. If there is generalizable knowledge in circuit representation and synthesis strategies, it should try to improve the generalization of the learned policy by using more data/better algorithms. If this problem in nature is not generalizable or learnable, then it is not necessary to use an RL agent to learn the synthesis strategy at the beginning. The proposed method does not fundamentally explain or solve the learnability of circuit synthesis problems, but rather uses two model ensembles to just cover some in-distribution data with learned model and out-of-distribution examples by search.

**Questions:**

Basically, it is listed in the weaknesses.

---

> ### Author Response · Authors · 2023-11-17
> **Official comment to qBwB**
>
> We thank the reviewer for appreciating our motivational observation that in practice benchmarks have large diversity and learned policy sometimes do not help for entirely novel circuits. The method also smoothly controls the recommendation from pre-trained agents while performing MCTS for synthesis recipe generation yielding runtime benefits. We address the reviewers' concerns below.
>
>
> Q1) *The overall framework basically does test-time augmentation. It assumes pre-trained policy is not generalizable to novel circuits, and by comparing the test example with the training example, it dynamically selects among two search strategies, but in a smooth way. The circuit similarity is a performance proxy for pre-trained agent and MCTS method, and uses that proxy to predict the weights to ensemble two models. The novelty, in this sense, is limited. Is it possible to combine more synthesis strategies based on a more general performance predictor at test time?*
>
>
> **Response:**
>
> We thank the reviewer for the question. With regards to novelty, as far as we are aware, adaptively mixing learning and search using an based on retrieval-guided cosine similarity is a new idea in literature that empirically outperforms several state-of-art methods, while a simpler solution without the judicious choice of how much learning to use (MCTS+L) does not. As such, our contributions include the algorithm to weight the pre-trained and pure search agents, how to set its hyper-parameters, and other policy network architecture contributions that are not in prior work and are key to the performance of our method.
>
> The reviewer makes an interesting point about ensembling other synthesis strategies. We agree that it might be possible to train a test time predictor that selects within this broader class of strategies, for example, SA+Pred etc. This would be an interesting avenue for future work. We do want to highlight one key point here: our pre-trained and MCTS search agents are not ensembles in the traditional sense: in fact, the pre-trained agent is trained to emulate MCTS search (as in alphaGO etc.), thus providing it a performance boost. As such, one could imagine pre-trained agents used to emulate other online search methods like simulated annealing, thus giving these methods a boost as well. Finally, these learning-augmented-search methods could be ensembled in the more traditional sense.
>
> Q2) *The usage of BERT is not well justified. There are other simpler methods to encode variable-length sequences, e.g., RNN. The attention model is also data-hungry during training. Why BERT is the most suitable encoder?*
>
> **Response:**
>
> Thank you for the suggestion; a comparison with an RNN (or LSTM) encoder is a good idea. To test this idea, we replaced our BERT-based encoder with an LSTM encoder and retrained our model. Our detailed results in ***Appendix D.5***. We find that our BERT-based encoder outperforms the LSTM-based encoder. Specifically, ABC-RL using an LSTM recipe encoder achieves **22.6%** geo-mean ADP reduction compared to **25.3%** using the existing BERT-based recipe encoder. The table is reproduced below for your convenience.
>
> | Recipe encoder | C1   | C2   | C3   | C4   | C5   | C6   | C7   | C8   | C9   | C10  | C11  | C12  | A1   | A2   | A3   | A4   | R1   | R2   | R3   | R4   | Geo. mean |
> |----------------|------|------|------|------|------|------|------|------|------|------|------|------|------|------|------|------|------|------|------|------|-----------|
> | Naive          | 17.0 | 16.5 | 14.9 | 13.1 | 44.6 | 16.9 | 10.0 | 23.5 | 16.3 | 18.8 | 10.6 | 19.0 | 36.9 | 51.7 | 9.9  | 20.0 | 15.5 | 23.0 | 28.0 | 26.9 | 21.2      |
> | LSTM-based     | 19.5 | 17.8 | 14.9 | 13.3 | 44.5 | 17.8 | 10.3 | 23.5 | 18.8 | 20.1 | 10.6 | 19.6 | 36.9 | 53.4 | 10.9 | 22.4 | 16.8 | 24.3 | 32.3 | 28.7 | 22.6      |
> | BERT-based     | 19.9 | 19.6 | 16.8 | 15.0 | 46.9 | 19.1 | 12.1 | 24.3 | 21.3 | 21.1 | 13.6 | 21.6 | 36.9 | 56.2 | 14.0 | 23.8 | 19.8 | 30.2 | 38.9 | 30.0 | 25.3      |

---

> ### Author Response · Authors · 2023-11-17
> **Official response to Reviewer qBwB (Part 2)**
>
> Q3) *The assumption that learned policy cannot generalize to diverse benchmarks is not well supported. If there is generalizable knowledge in circuit representation and synthesis strategies, it should try to improve the generalization of the learned policy by using more data/better algorithms. If this problem in nature is not generalizable or learnable, then it is not necessary to use an RL agent to learn the synthesis strategy at the beginning. The proposed method does not fundamentally explain or solve the learnability of circuit synthesis problems, but rather uses two model ensembles to just cover some in-distribution data with learned model and out-of-distribution examples by search.*
>
> **Response:**
>
> We thank the reviewer for raising this point. We would like to emphasize that ABC-RL never defaults to a pure search solution, but instead chooses the right mix of learning and search on a per-benchmark basis. Emprically, across our test circuits, ABC-RL never sets $\alpha=1$, i.e., never fully turns off the learned agent, thus ensuring that the learned policy is constantly helping, albeit by different amounts. This is evidenced by the fact that ABC-RL outperforms pure MCTS search on all circuits except on one (for which both results are the same). Therefore, in each of these instances, the learned agent is actually generalizing to test inputs and helping to boost performance over the pure search agent, but by different amounts. In sum, our claim is not that learning does not help (in fact, on average even MCTS+L is better than MCTS), but that the amount of learning should be judiciously selected using our proposed scheme.
>
> Finally, one can imagine a test circuit that is so novel that ABC+RL sets $\alpha=1$ in which case we would default to search. However, in our view, such an “out-of-distribution” (OOD) circuit by itself would not mean that learning is not useful, given that OOD inputs are routinely encountered across a range of ML applications. Nevertheless, we note that ABC-RL itself does not label test circuits explicitly as either in-distribution or OOD.

---

> > ### Author Response · Authors · 2023-11-22
> > **Highlighting a new result on generalizability, and any remaining questions we can address?**
> >
> > We thank the reviewer again for their insightful questions. As the rebuttal period comes to a close, we are reaching out to highlight a new experimental result obtained in response to Reviewer 3ENJ that speak directly to your question about generalizability. Specifically, we show ABC-RL’s ability to generalize to entirely new, unseen circuits from OpenCores, which are not part of the three standard logic synthesis benchmark suites that we used.
> >
> > **Generalization capability of ABC-RL**: As also asked by reviewer 3ENJ, we evaluated ABC-RL on 9 circuits extracted out of Opencore circuits which were never seen before. We report area-delay reduction (in %) obtained by ABC-RL and compare it with MCTS:
> >
> > | Method | fpu   |  ac97_ctrl  | fir  | iir   | aes  | wb_dma  |
> > |----------------|------|------|------|------|------|------|
> > | MCTS         | 11.61 | 29.80 | 15.21 | 4.67 | 51.62 | 16.61 |
> > | ABC-RL     | 16.86 | 34.35 | 17.46 | 6.09 | 72.36 | 27.55 |
> >
> >
> > Second, we wanted to highlight that we have also justified our use of BERT via new experiments summarized below for easy reference.
> >
> > **Choice of BERT-based synthesis recipe encoder**: As asked, we performed additional experiment replacing BERT-based encoder with a simple LSTM-based encoder. *ABC-RL using an LSTM recipe encoder achieves 22.6% geo-mean ADP reduction compared to 25.3% using the existing BERT-based recipe encoder.*  We presented detailed experimental results in our response.
> >
> > | Recipe encoder | C1   | C2   | C3   | C4   | C5   | C6   | C7   | C8   | C9   | C10  | C11  | C12  | A1   | A2   | A3   | A4   | R1   | R2   | R3   | R4   | Geo. mean |
> > |----------------|------|------|------|------|------|------|------|------|------|------|------|------|------|------|------|------|------|------|------|------|-----------|
> > | Naive          | 17.0 | 16.5 | 14.9 | 13.1 | 44.6 | 16.9 | 10.0 | 23.5 | 16.3 | 18.8 | 10.6 | 19.0 | 36.9 | 51.7 | 9.9  | 20.0 | 15.5 | 23.0 | 28.0 | 26.9 | 21.2      |
> > | LSTM-based     | 19.5 | 17.8 | 14.9 | 13.3 | 44.5 | 17.8 | 10.3 | 23.5 | 18.8 | 20.1 | 10.6 | 19.6 | 36.9 | 53.4 | 10.9 | 22.4 | 16.8 | 24.3 | 32.3 | 28.7 | 22.6      |
> > | BERT-based     | 19.9 | 19.6 | 16.8 | 15.0 | 46.9 | 19.1 | 12.1 | 24.3 | 21.3 | 21.1 | 13.6 | 21.6 | 36.9 | 56.2 | 14.0 | 23.8 | 19.8 | 30.2 | 38.9 | 30.0 | 25.3      |

---

### Official Review · Reviewer_3ENJ · 2023-10-31

**Soundness:** 2 fair
**Presentation:** 3 good
**Contribution:** 2 fair
**Rating:** 5
**Confidence:** 4

**Summary:**

This paper presents ABC-RL, a retrieval-guided RL approach to generate an optimized synthesis recipe for logic synthesis. ABC-RL tunes a weight that adeptly adjusts recommendations from pre-trained agents during testing. Given a circuit with high similarity to the circuits in the training dataset, ABC-RL assigns high weight to the policy by the RL agent. Otherwise, ABC-RL tends to rely more on the default searching strategy.

**Strengths:**

1.	This paper introduces a retrieval-guided RL agent to search for an optimized synthesis recipe for logic synthesis. ABC-RL selectively leverages the knowledge obtained during training based on the similarity between training and testing samples. This approach can address the issue of RL performance decrease caused by different data distributions.

2.	The authors claim that ABC-RL can achieve up to 24.8% QoR improvement and reduce runtime up to 9x.

**Weaknesses:**

1.	ABC-RL calculates the cosine similarity between the embeddings of training and testing samples to determine the similarity score. Therefore, the quality of AIG embeddings is the key to the entire methodology. Unfortunately, the authors do not explore this issue and simply use a 3-layer GCN to do it. It's hard to believe such an approach could work properly. A circuit graph is a lot more complicated than a plain graph, not only it's directed, but more importantly, it has unique functions associated with it. Two AIGs could be very similar in terms of structure but differ significantly in terms of functionalities. Consequently, they may require different synthesis recipes, isn't it?

2.	The experiments are conducted with a very small dataset, which only includes 23 netlists for training. This is not convincing. The proposed agent should have seen sufficient circuit designs to come up with a good RL strategy.

**Questions:**

Please refer to the weakness part.

---

> ### Author Response · Authors · 2023-11-17
> **Official response to reviewer 3ENJ**
>
> We thank the reviewer for appreciating ABC-tuning strategy to adjust recommendations from pre-trained agents during test time and address the issue of learned policies sometimes underperforming pure search. We address the reviewer’s primary concerns below.
>
>
> Q1) *Quality of  AIG embeddings: The reviewer is concerned about usage of 3-layer GCN network to encode AIG graphs which are directed and represent functionality. Thus, two AIG’s with the same structure can differ in functionality and hence require different synthesis recipes. However, our 3-layer GCN may fail to distinguish between such similar AIG structure but different functionality.*
>
> **Response:**
>
> Thank you for the question. We note that unlike CNNs, GNN/GCN architectures tend to be shallower because deeper networks suffer from a well-studied “over-smoothing” [1,2] effect; i.e., as GCN depth increases each node’s features depend on all other nodes, resulting in an ``averaging” effect that causes all nodes to have similar embeddings. Therefore, GNNs often achieve optimal classification performance when networks are shallow, and many widely used GNNs/GCNs architectures are no deeper than 4 layers [2,3,4]. Prior work on ML for logic synthesis, for instance, the Online-RL approach (Zhu et al. 2020), and SA+Pred. (Chowdhury et al., 2022) methods we compare against also use GCNs with up to 4 layers. Mirhoseini et al, (2021) use only a 2-layer GCN in their RL agent for chip placement. As such, we would like to emphasize that our GCN architecture depth is consistent with common practice in the graph learning literature.
>
>
> [1] Li, Qimai et al. Deeper insights into graph convolutional networks for semi-supervised learning. In Thirty-Second AAAI Conference on Artificial Intelligence, 2018.
>
> [2] Wu, Xinyi, et al. "A Non-Asymptotic Analysis of Oversmoothing in Graph Neural Networks." in International Conference on Learning Representations (ICLR) 2022.
>
> [3] Errica, F., ert al. A Fair Comparison of Graph Neural Networks for Graph Classification. in International Conference on Learning Representations (ICLR) 2020.
>
> [4] Wu, Zonghan et al. A comprehensive survey on graph neural networks. in IEEE transactions on neural networks and learning systems 32.1 (2020): 4-24.
>
> Q2) *The experiments are conducted with a very small dataset, which only includes 23 netlists for training. This is not convincing. The proposed agent should have seen sufficient circuit designs to come up with a good RL strategy.*
>
> **Response:**
>
> We note that although we use 23 netlists for training, the actual number of data samples that our agent is trained on is much larger, i.e., about 11,500 different states that are generated during the episodes of training. In each episode, 230 new graphs are generated using most promising synthesis recipes using MCTS, and since we train for 50 episodes, that yields our final tally of 11,500.
>
> Recent work in RL for hardware optimization have similar number of training examples: 20 netlists in [4] and 12 in [5]. In part, the limitation stems from the size of publicly available datasets. In our paper, we have tried to address this issue by combining both commonly used public datasets for logic synthesis, MCNC (1991) and EPFL benchmarks (2015), yielding our eventual tally of 23 training netlists.
>
> [4] Mirhoseini, Azalia, et al. "A graph placement methodology for fast chip design." Nature 2021
>
> [5] Lai, Yao, et al. "Chipformer: Transferable chip placement via offline decision transformer." International Conference on Machine Learning 2023

---

> > ### Comment · Reviewer_3ENJ · 2023-11-19
> >
> > Thanks to the authors for the answers. However, they do not address my questions well.
> >
> > 1. Regarding the AIG embedding, the problem is not about the number of layers in the GCN. Circuits are directed graphs with built-in functionalities. Using a simple GCN for AIG embedding, both the structural information and the functional information suffer from great loss. Consequently, it is likely that two vastly different circuits generate similar GCN embedding with this work and hence use similar synthesis recipes, but they shouldn't.
> >
> > 2. Indeed, we don't have lots of publicly available datasets that are readily applicable to this work. On the one hand, the authors should discuss its implications, i.e., what kind of circuits may benefit from this solution. On the other hand, the authors can extract more circuits from OpenCore or GitHub projects to make the work more convincing. I understand this would involve lots of work, but the authors should at least test a few circuits that are outside of the benchmark dataset to demonstrate the generalization capability of the proposed solution.

---

> ### Author Response · Authors · 2023-11-19
> **Response for reviewer 3ENJ (Part 2)**
>
> We acknowledge the reviewer's diligence in pointing out that our rebuttal response may not have comprehensively addressed his inquiries. We address the concerns below:
>
> *Q1) Regarding the AIG embedding, the problem is not about the number of layers in the GCN. Circuits are directed graphs with built-in functionalities. Using a simple GCN for AIG embedding, both the structural information and the functional information suffer from great loss. Consequently, it is likely that two vastly different circuits generate similar GCN embedding with this work and hence use similar synthesis recipes, but they shouldn't.*
>
> **Response**:
>
> We articulate our rationale for utilizing a simple Graph Convolutional Network (GCN) architecture to encode AIGs for the generation of synthesis recipes aimed at optimizing the area-delay product. We elucidate why this approach is effective and support our argument with an experiment that validates its efficacy:
>
> * **Working principle of logic synthesis transformations**: Logic synthesis transformations of ABC (and in general commercial logic synthesis tools) including *rewrite*, *refactor* and *re-substitute* performs transformations at a local subgraph levels rather than the whole AIG structure. For e.g. *rewrite* performs a backward pass from primary outputs to primary inputs, perform k-way cut at each AIG node and replace the functionality with optimized implementation available in the truth table library of ABC. Similarly, *refactor* randomly picks a fan-in cone of an intermediate node of AIG and replaces it with a different implementation if it reduced the nodes of AIG. Thus, effectiveness of any synthesis transformations  do not require deeper GCN layers; capturing neighborhood information upto depth 3 in our case worked well to extract features out of AIG which can help predict which transformation next can help reduce area-delay product.
>
> * **Feature initialization of nodes in AIG**: The node in our AIG encompasses two important feature: i) Type of node (Primary Input, Primary Output and Internal Node) and ii) Number of negated fan-ins. Thus, our feature initialization scheme takes care of the functionality even though the structure of AIG are exactly similar and the functionality. Thus, two AIG having exact same structure but edge types are different (dotted edge represent negation and solid edge represent buffer), the initial node features of AIG itself will be vastly different and thus our 3-layer GCNs have been capable enough to distinguish between them and generate different synthesis recipes.
>
> To illustrate this point further, we conduct a concise experiment involving a test circuit, 'div.' In this experiment, we generate additional AIGs by randomly flipping the edges of the AIG structure, thereby creating structural variants with distinct functionalities. For example, when we mention '10%,' it signifies that 10% of the edge types in the AIG were flipped. We employ the equivalence check feature of the ABC tool to ascertain whether these newly generated AIGs are structurally identical but functionally divergent. Additionally, we present the similarity score (cosine distance) of these AIGs with respect to the original circuit and the achieved Area-Delay Product (ADP) reduction obtained through Monte Carlo Tree Search (MCTS) and ABC-RL.
>
> | Circuit (div) | Similarity score  | Equivalence check   | MCTS   | ABC-RL |
> |---------------------|------|------|------|------|
> | orig                  | 0.0 | PASS | 46.00 | 56.2 |
> | 10% flipped     | 0.029 | FAIL | 42.36 | 49.99 |
> | 20% flipped     | 0.066 | FAIL | 62.33 | 64.11 |
> | 30% flipped     | 0.145 | FAIL | 57.32 | 57.33|
> | 40% flipped     | 0.239 | FAIL |  41.36| 42.01|
> | 50% flipped     | 0.568 | FAIL | 39.15 | 40.68|
>
> Our results highlight that the feature extracted by our 3-layer GCN architecture is able to capture functional differences having same skeleton AIG structure resulting in different similarity scores and ABC-RL performs better than MCTS.

---

> > ### Comment · Reviewer_3ENJ · 2023-11-19
> >
> > Thanks to the authors for the additional experiments, but again, it is not convincing. First of all, if "flipping the edges" means exchanges between NOT and AND gate types, wouldn't it cause illegal circuits? Secondly, how can you say that changing 10% of the gate types as two structurally identical AIGs?

---

> > > ### Author Response · Authors · 2023-11-20
> > > **Clarifying the AIG representation**
> > >
> > > We thank the reviewer for their thoughtful feedback. To answer the reviewer's concern, we emphasize that in our flipping experiment, we are **not** replacing NOTs with ANDs (or vice-versa). Flipping only replaces wires with NOT gates, or NOT gates with wires. Thus the resulting circuits are always legal: all gates in the resulting circuit have valid inputs and outputs, and the circuit is still acyclic and directed. The modified AIG does have different Boolean functionality (as intended), and as verified via simulation.
> > >
> > > To further clarify, we use the same AIG format introduced in [1,2] and commonly used in literature: nodes in the AIG represent AND gates, Primary Inputs or Primary Outputs. On the other hand, NOT gates are represented by edges: dashed edges represent NOT gates (i.e., the output of the edge is a logical negation of its input) and solid edges represent a simple wire whose output equals its input.
> > >
> > > From a structural standpoint, flipping edges does not change the connectivity of the AIG or its structural properties like node depth. As such, in the experiment we ran, we were trying to see if our GCN representation could catch differences in AIGs that are functionally different but whose structure is as similar as possible.
> > >
> > > We hope this clarifies our experimental setup above, and apologize for any confusion or lack of clarity.
> > >
> > > [1] Mischenko et al.FRAIGs: A Unifying Representation for Logic Synthesis and Verification",  ERL Technical Report, 2005
> > > [2] Kuehlmann, Andreas, et al. "Robust Boolean reasoning for equivalence checking and functional property verification." IEEE Transactions on Computer-Aided Design of Integrated Circuits and Systems 21.12 (2002): 1377-1394

---

> > > > ### Comment · Reviewer_3ENJ · 2023-11-21
> > > >
> > > > In your previous rebuttal, you mention the AIG as: "The node in our AIG encompasses two important feature: i) Type of node (Primary Input, Primary Output and Internal Node) and ii) Number of negated fan-ins. " In the main text of the paper, it is written as: "The AIG has only AND gates and NOT as edges ..." In the current rebuttal, it is said that there are actually two types of edges. This is really confusing, and it is essential to make it clear in the main text of the paper. Moreover, under such circumstances, you need to elaborate more on the GNN aggregation procedure because the regular GCN doesn't support different types of edges in aggregation. Do you use separate channels for different types of edges?
> > > >
> > > > As for the question itself, I cannot accept the claim that replacing many NOT gates in the circuit with wires and vice versa still holds structural similarity.
> > > >
> > > > In fact, there have been many advancements in circuit embedding techniques in recent years, e.g., the DeepGate family.

---

> > > > > ### Author Response · Authors · 2023-11-21
> > > > > **Clarifications**
> > > > >
> > > > > *Q1) In your previous rebuttal, you mention the AIG as: "The node in our AIG encompasses two important feature: i) Type of node (Primary Input, Primary Output and Internal Node) and ii) Number of negated fan-ins. " In the main text of the paper, it is written as: "The AIG has only AND gates and NOT as edges ..." In the current rebuttal, it is said that there are actually two types of edges. This is really confusing, and it is essential to make it clear in the main text of the paper.*
> > > > >
> > > > > **Response**:
> > > > >
> > > > > We see where the confusion is coming from, and apologize for not being clearer in our responses. Thank you for the opportunity to let us clarify below.
> > > > >
> > > > > In our implementation, we are dealing with two types of AIG formats: the standard format used by the ABC tool (lets call that the external format); and the internal AIG format/representation we use as input to the GCN. In our rebuttal, we were switching between the two depending on context without mentioning which, thus creating the confusion.
> > > > >
> > > > > Below, we first explicitly clarify our responses:
> > > > >
> > > > > *"The node in our AIG encompasses two important feature: i) Type of node (Primary Input, Primary Output and Internal Node) and ii) Number of negated fan-ins.”*
> > > > >
> > > > > **Response**: Here we were referring to our internal AIG representation within the GCN. This representation, used in a line of prior work on logic synthesis [3,4,5,6], uses a two dimensional vector to encode node-level features: (i) node type (AND, PI, PO) and (ii) number of dashed (or negated) fan-in edges at the node. We have updated the paper with a description of our internal representation in Appendix B.1 where we discuss our GCN implementation.
> > > > >
> > > > > *In the main text of the paper, it is written as: "The AIG has only AND gates and NOT as edges ..." In the current rebuttal, it is said that there are actually two types of edges. This is really confusing, and it is essential to make it clear in the main text of the paper.*
> > > > >
> > > > > **Response**: Here we were referring to the external AIG format. In main draft, when we said "The AIG has only AND gates and NOT as edges ..." we were describing ABC’s standard external AIG format. This sentence is indeed confusing. We have clarified it in the new draft as follows: "The AIG represents AND gates as nodes, wires/NOT gates as solid/dashed edges…"  The external AIG representation.
> > > > >
> > > > > Our comment that edges have two types was for the external format, since we were discussing the flipping experiment.
> > > > >
> > > > > We hope that the context is now clear, and apologize again for the confusion. We have updated the paper to clearly describe each.
> > > > >
> > > > > *Q2) Moreover, under such circumstances, you need to elaborate more on the GNN aggregation procedure because the regular GCN doesn't support different types of edges in aggregation. Do you use separate channels for different types of edges?*
> > > > >
> > > > > **Response**: We agree, and have clarified the details of the internal AIG representation used in our GCN implementation in Appendix B.1. As noted by the reviewer, we use a standard GCN and a separate channel to encode the number of dashed/inverted edges at the fan-in of each node.
> > > > >
> > > > > *Q3) As for the question itself, I cannot accept the claim that replacing many NOT gates in the circuit with wires and vice versa still holds structural similarity.*
> > > > >
> > > > > **Response**: Our flipping experiment was in response to the reviewer’s original question: *"Thus, two AIG’s with the **same structure can differ in functionality** and hence require different synthesis recipes. However, our 3-layer GCN may fail to distinguish between such similar AIG structure but different functionality."*
> > > > >
> > > > > By only replacing a small fraction of the NOTs in the circuit with wires (or vice-versa) we were trying to minimally perturb circuit structure, which we perceived as the logic independent features of a circuit. As such, viewing a wire as a buffer, a buffer’s logic either converts to a NOT or vice-versa. But, we would be happy to try other transformations (given time constraints) if the reviewer could suggest other transformations that preserve structure while modifying functionality.
> > > > >
> > > > > *Q4) In fact, there have been many advancements in circuit embedding techniques in recent years, e.g., the DeepGate family.*
> > > > >
> > > > >
> > > > > **Response**: The reviewer makes an excellent point; for instance, pre-trained DeepGate ([1,2]) concatened with our embeddings (with/without finetuning) might further improve our results. This would be an excellent avenue for future research, and we added a comment to this effect in the Related Work section of the updated draft. Note that based on the reviewer’s previous comments, we have also already launched a 6-layer GCN experiment, which is still training.

---

> > > > > > ### Author Response · Authors · 2023-11-21
> > > > > > **References for previous clarifications**
> > > > > >
> > > > > > **References**
> > > > > >
> > > > > > [1] Li, Min, et al. "Deepgate: Learning neural representations of logic gates." Proceedings of the 59th ACM/IEEE Design Automation Conference. 2022.
> > > > > >
> > > > > > [2] Shi, Zhengyuan, et al. "DeepGate2: Functionality-Aware Circuit Representation Learning." arXiv preprint arXiv:2305.16373 (2023).
> > > > > >
> > > > > > [3] Zhu, Keren, et al. "Exploring logic optimizations with reinforcement learning and graph convolutional network." Proceedings of the 2020 ACM/IEEE Workshop on Machine Learning for CAD. 2020.
> > > > > >
> > > > > > [4] Chowdhury, Animesh Basak, et al. "Bulls-Eye: Active Few-shot Learning Guided Logic Synthesis." IEEE Transactions on Computer-Aided Design of Integrated Circuits and Systems (2022).
> > > > > >
> > > > > > [5] Yang, Chenghao, et al. "Logic synthesis optimization sequence tuning using RL-based LSTM and graph isomorphism network." IEEE Transactions on Circuits and Systems II: Express Briefs 69.8 (2022): 3600-3604.
> > > > > >
> > > > > > [6] Chowdhury, Animesh Basak, et al. "OpenABC-D: A large-scale dataset for machine learning guided integrated circuit synthesis." arXiv preprint arXiv:2110.11292 (2021)

---

> > > > > > > ### Author Response · Authors · 2023-11-21
> > > > > > > **Summary**
> > > > > > >
> > > > > > > As a summary, we would like to emphasize a few points:
> > > > > > > * For this paper, ***we chose to use the same GCN embeddings as prior work on logic synthesis ([3,4,5,6]) since our key novelty was not on the GCN architecture***, but on learning from historical data via retrieval-guided $\alpha$ tuning, and logic-synthesis specific architecture changes like our recipe encoder.
> > > > > > >
> > > > > > > * Embeddings like DeepGate supervise on tasks like predicting signal probabilities, seeking “general” embeddings. ***Our embeddings are directly supervised on the task we seek to perform: logic synthesis***. As such, we expect our embeddings to pick up aspects of function and structure relevant for logic synthesis, but not necessarily all aspects of functionality since that is not our end goal.
> > > > > > >
> > > > > > > * Nevertheless, in response to the reviewer’s excellent questions, we have shown that ***ABC-RL embeddings do pick up aspects of circuit function***, since (i) we are able to retrieve functionally similar circuits from the training set, (ii) our embeddings can differentiate modified functionality via “flipping,” (notwithstanding whether flipping preserves structure or not).
> > > > > > >
> > > > > > > * Finally, also in response to the reviewer’s incisive questions, we also showed ***ABC-RL generalizes to even OpenCores circuits***, despite these not being seen in the past at all.
> > > > > > >
> > > > > > > Going back to the reviewer’s original concerns, our embeddings do appear to pick up key aspects of circuit functionality, based on the new experiments we have conducted in response to the reviewer. We agree that the AIG embeddings could be further improved, but emphasize that these improvements would be orthogonal to the main goal of our paper which is retrieval-guided RL. Thus, we have focused on showing significant improvements in SOTA obtained from these new contributions, while using embeddings from leading prior work [3,4,5,6] that we compare against.
> > > > > > >
> > > > > > > **References**:
> > > > > > >
> > > > > > > [1] Li, Min, et al. "Deepgate: Learning neural representations of logic gates." Proceedings of the 59th ACM/IEEE Design Automation Conference. 2022.
> > > > > > >
> > > > > > > [2] Shi, Zhengyuan, et al. "DeepGate2: Functionality-Aware Circuit Representation Learning." arXiv preprint arXiv:2305.16373 (2023).
> > > > > > >
> > > > > > > [3] Zhu, Keren, et al. "Exploring logic optimizations with reinforcement learning and graph convolutional network." Proceedings of the 2020 ACM/IEEE Workshop on Machine Learning for CAD. 2020.
> > > > > > >
> > > > > > > [4] Chowdhury, Animesh Basak, et al. "Bulls-Eye: Active Few-shot Learning Guided Logic Synthesis." IEEE Transactions on Computer-Aided Design of Integrated Circuits and Systems (2022).
> > > > > > >
> > > > > > > [5] Yang, Chenghao, et al. "Logic synthesis optimization sequence tuning using RL-based LSTM and graph isomorphism network." IEEE Transactions on Circuits and Systems II: Express Briefs 69.8 (2022): 3600-3604.
> > > > > > >
> > > > > > > [6] Chowdhury, Animesh Basak, et al. "OpenABC-D: A large-scale dataset for machine learning guided integrated circuit synthesis." arXiv preprint arXiv:2110.11292 (2021)

---

> ### Author Response · Authors · 2023-11-19
> **Response for reviewer 3ENJ (Part 3)**
>
> *Q2) Indeed, we don't have lots of publicly available datasets that are readily applicable to this work. On the one hand, the authors should discuss its implications, i.e., what kind of circuits may benefit from this solution. On the other hand, the authors can extract more circuits from OpenCore or GitHub projects to make the work more convincing. I understand this would involve lots of work, but the authors should at least test a few circuits that are outside of the benchmark dataset to demonstrate the generalization capability of the proposed solution.*
>
> **Response**:
>
> We appreciate the reviewer's understanding that extracting the circuits from Opencore or Github projects and training our framework require considerable more time to demonstrate further effectiveness (Our training time on 23 circuits with parallelized implementation takes around 9 days). However, to the best possible extent we provide results on 9 circuits extracted out of Opencore with our ABC-RL model trained on 23 circuits. We report area-delay reduction (in %) obtained by ABC-RL and compare it with MCTS:
>
> | Method | fpu   |  ac97_ctrl  | fir  | iir   | aes  | wb_dma  |
> |----------------|------|------|------|------|------|------|
> | MCTS         | 11.61 | 29.80 | 15.21 | 4.67 | 51.62 | 16.61 |
> | ABC-RL     | 16.86 | 34.35 | 17.46 | 6.09 | 72.36 | 27.55 |
>
> We also want to highlight the fact that: in hardware domain the distribution study of circuits in terms of functionality and structure is still an open problem to explore. For the kind of circuits that can benefit well from ABC-RL (i.e. circuits on which ABC-RL function well and generalize), we observe our cosine distance capture this notion well during test time.
>
> Also asked by reviewer yxFf, we presented  this data in ***Appendix D.4 (Table 6)***. Several interesting observations we have drawn from it: first, note that the closest circuits in the training set are often semantically similar (*alu2 for alu4, apex3 for apex1, sqrt for multiplier*), suggesting that the learned embeddings are capturing netlist structure/function. Second, netlists for which we found MCTS+L to underperform MCTS alone, for example, square and cavlc have relatively **high** cosine distances of 0.012 and 0.007, respectively, relative to the $\delta_{th}=0.007$ in Equation 4.

---

> > ### Comment · Reviewer_3ENJ · 2023-11-19
> >
> > I thank the authors for doing the experiments on Opencore circuits, and I've raised my score for answering this question.

---

### Official Review · Reviewer_Pwwd · 2023-11-01

**Soundness:** 3 good
**Presentation:** 3 good
**Contribution:** 2 fair
**Rating:** 6
**Confidence:** 3

**Summary:**

This work introduces, ABC-RL, an MCTS+Learning algorithm to optimize the transformation recipes in logic synthesis to minimize the area delay product (ADP).  It employs GCN for AIG features and a transformer for applied transformations. It introduces a hyperparameter to adjust recommendations from pre-trained agents during the search. The hyperparameter is determined by similarity computed from GNN features learned during training. The key observation it leverages from hardware designs is that they contain both familiar and entirely new components. The empirical results show ABC-RL improves synthesized circuit Quality-of-Result (QoR) by up to 24.8% over SOTA methods and provides an average runtime speed-up of 1.6× compared to baseline MCTS.

**Strengths:**

1. The policy network architecture for recipe encoding and AIG embedding is reasonably selected.
2. The novel introduction of a similarity-score-based hyperparameter effectively enhances ABC-RL convergence, as observed in the results.
3. The study provides a comprehensive evaluation, comparing various search algorithms for logic synthesis recipe optimization, including MCTS, online RL, and simulated annealing. It also includes comparisons with MCTS+Learning with fine-tuning.

**Weaknesses:**

1. It would be beneficial to provide an estimate of the number of gates in each benchmark circuit. One concern with this approach is whether GCN-based AIG embeddings can effectively scale to real-world circuit designs.
2. With the integration of GNN + transformer features. It is likely that the compute complexity and runtime of each search iteration would be higher than the baseline MCTS.

**Questions:**

1. How is it compared to non-learning-based recipes? Is there an O3 flag for logic synthesis?
2. How does its wall clock time (not iterations) compare to other algorithms such as MCTS and SA+Pred?

---

> ### Author Response · Authors · 2023-11-17
> **Official response to reviewer Pwwd**
>
> We thank the reviewer for their insightful feedback and address the concerns raised:
>
> Q1) *It would be beneficial to provide an estimate of the number of gates in each benchmark circuit. One concern with this approach is whether GCN-based AIG embeddings can effectively scale to real-world circuit designs.*
>
> **Response:**
>
> Thank you for the suggestion. We provide detailed characterization in ***Appendix C.2*** listing all the circuits from MCNC and EPFL benchmarks about primary inputs and outputs, number of nodes and level of AIG graphs. The benchmarks circuits range from 400-46,000 nodes. In terms of applicability to real-world designs, we note that logic synthesis is often performed hierarchically because Verilog code itself is written in a modular fashion and because of the large runtimes of *flattened* synthesis. As such, our results (and indeed those in the logic synthesis literature that all make use of IWLS/EPFL) can be viewed as synthesis runs on leaf and intermediate nodes of a hierarchical flow.
>
> *With the integration of GNN + transformer features. It is likely that the compute complexity and runtime of each search iteration would be higher than the baseline MCTS. How does its wall clock time (not iterations) compare to other algorithms such as MCTS and SA+Pred?*
>
> **Response:**
>
> This is an excellent point. We note that over iterations of MCTS, SA+Pred or ABC-RL search it is the actual synthesis of logic circuits using sample recipes that dominates the runtime, leaving GNN inference as a relatively negligible component to wall-clock time. Therefore, the overhead of ABC-RL over MCTS in terms of wall-clock time is only **1.5% (geo. mean)** as shown in ***Appendix D.6 (Table 8)***. Our **wall-clock speed-up** over MCTS at iso-QoR is **3.75x**.
>
>
> Q3) *How is it compared to non-learning-based recipes? Is there an O3 flag for logic synthesis?*
>
> **Response:**
>
> The publicly known non-learned recipe for the ABC synthesis tool is resyn2 which is heuristically tailored to optimize for area-delay-product (ADP), our QoR metric. As such it can be viewed as akin to an -O3 flag for C/C++. For this reason, all our reported results in ADP improvements are on top of  (i.e., relative to) the resyn2 recipe. Over time, experienced EDA engineers might develop their own intuitions as to which recipes work best for different circuits, but we are not aware of any formal documentation to this effect.

---

> > ### Author Response · Authors · 2023-11-22
> > **Any remaining questions we can address?**
> >
> > We thank the reviewer again for their insightful questions. As the rebuttal period comes to a close, we are reaching out to see if our experimental results and data in response to your questions have addressed your concerns. As a summary we, (i) provided data on circuit sizes as requested; (ii) performed requested wall-clock runtime comparisons and demonstrated that ABC-RL has only 1.5% wall-clock overhead; and (iii) discussed the -O3 flag equivalent for logic synthesis. We would be happy to address any other concerns in the time left.

---

### Author Response · Authors · 2023-11-17
**Official response**

We thank the reviewers for their thoughtful comments, and the opportunity to revise the paper.

In our response, we have performed new experiments and provided several new data points in addition to quantitative and qualitative responses to the reviewers’ questions.

We look forward to engaging further should the reviewers have additional questions or concerns.

**New experiments:**

*1: Comparison with DRiLLS [1]*

*2: RNN/LSTM based recipe encoders instead of BERT encoders*

**Additional datapoints:**

*1: Performance on training and validation data*

*2: Nearest neighbor data for test circuits*

*3: Wall-clock runtime comparisons*

[1] Hosny, Abdelrahman, et al. "DRiLLS: Deep reinforcement learning for logic synthesis." 2020 25th Asia and South Pacific Design Automation Conference (ASP-DAC). IEEE, 2020.

---

### Meta-Review · Area_Chair_e7nN · 2023-12-06

**Metareview:**

The authors propose an approach for logic synthesis that leverages pre-trained agents from an initial training corpus using reinforcement learning, and adjusts recommendations from these agents at test time through parameter tuning. To adjust the parameters, this method uses cosine similarity between graph embeddings of the input test AIG to AIGs seen during training time. Evaluations on a wide range of datasets from the logic synthesis community demonstrate better empirical performance compared to baselines. Reviewers agree that the evaluations are extensive and results are promising both in terms of QoR and run-time reduction. Additionally, some reviewers agree that the method is effective, tackles an important problem in logic synthesis, and is novel. The main concerns remaining are on the scalability of this method to real-world circuits as well as on the reliability of the AIG embeddings for retrieval at test time. Ultimately, I think the methodological contributions and strong experimental result is sufficient for acceptance.

**Justification For Why Not Higher Score:**

There remains concerns on scalability of the method to real-world circuits. Additionally, the model relies on the learned AIG embeddings from a GCN - there could be more experiments/discussions on the limitations & reliability of these embeddings.

**Justification For Why Not Lower Score:**

Reviewers appreciate the performance improvement of the proposed method relative to baseline, and extensive evaluations. Additionally, the method is well-motivated for performing logic synthesis. Many concerns from reviewers have been addressed during the author rebuttal period.

---

### Decision · Program_Chairs · 2024-01-16

Accept (poster)